# VideoArtGS: Building Digital Twins of Articulated Objects from Monocular Video

## Abstract

Building digital twins of articulated objects from monocular video presents an essential challenge in computer vision, which requires simultaneous reconstruction of object geometry, part segmentation, and articulation parameters from limited viewpoint inputs. Monocular video offers an attractive input format due to its simplicity and scalability; however, it's challenging to disentangle the object geometry and part dynamics with visual supervision alone, as the joint movement of the camera and parts leads to ill-posed estimation. While motion priors from pre-trained tracking models can alleviate the issue, how to effectively integrate them for articulation learning remains largely unexplored. To address this problem, we introduce VideoArtGS, a novel approach that reconstructs high-fidelity digital twins of articulated objects from monocular video. We propose a motion prior guidance pipeline that analyzes 3D tracks, filters noise, and provides reliable initialization of articulation parameters. We also design a hybrid center-grid part assignment module for articulation-based deformation fields that captures accurate part motion. VideoArtGS demonstrates state-of-the-art performance in articulation and mesh reconstruction, reducing the reconstruction error by about **two orders of magnitude** compared to existing methods. VideoArtGS enables practical digital twin creation from monocular video, establishing a new benchmark for video-based articulated object reconstruction. More visualized results are made publicly available at: https://videoartgs-2026.github.io.

## 1 Introduction

Articulated objects, prevalent in our daily life, are becoming a major focus in recent research for computer vision and robotics (Weng et al., 2024; Luo et al., 2025; Liu et al., 2024b; Deng et al., 2024; Yang et al., 2023; Liu et al., 2024a). Reconstructing interactable digital twins of articulated objects from visual observations is fundamental to advancing applications in augmented reality, robotics simulation, and interactive scene understanding. By generating digital twins from simple inputs like video, we can significantly accelerate the development of intelligent systems, particularly by bridging the sim-to-real gap for robotic manipulation and interaction tasks (Torne et al., 2024; Kerr et al., 2024). To build powerful and generalizable robotic systems, reconstructing interactable objects from monocular video represents a critical frontier, as this would unlock the ability to learn from the vast amount of videos available online and allow robots to model the world through their own eyes.

Recent approaches to reconstructing articulated objects can be broadly categorized into two families based on the way to estimate articulation parameters. One family employs a feed-forward model to predict articulation parameters directly (Mandi et al., 2024; Le et al., 2025; Jiang et al., 2022). These methods, however, struggle with scalability and generalization, as they require extensive training on annotated data, which often fails to transfer to novel, real-world settings. Creating datasets that comprehensively cover the sheer combinatorial complexity of real-world objects, articulation types, and viewing conditions is practically infeasible. A second, more common family reconstructs objects by explicitly estimating joint parameters from multi-view images of the object in two or more discrete states (Liu et al., 2025; 2023a; Weng et al., 2024; Lin et al., 2025; Yu et al., 2025). While these methods benefit from strong geometric constraints, they require controlled, often cumbersome, data capture setups that limit their use outside the lab. This approach is not only constrained by impractical data capture requirements but is also highly brittle; slight misalignments in the coordinate frames between states can cause catastrophic failures in prediction accuracy. A far more practical

and scalable paradigm is reconstructing articulated objects from casually captured monocular videos, which enables the ability to learn from internet videos and allows robotic agents to model objects directly from their visual observations.

However, the convenience of video capture introduces a profound technical challenge: the reconstruction problem becomes fundamentally ill-posed. From a single, moving viewpoint, the observed pixel motion results from four entangled factors: camera trajectory, object geometry, part segmentation, and articulation-based part dynamics. Disentangling these variables without the strong parallax cues from multi-view data is highly ambiguous. Consequently, prior video-based methods often produce distorted geometries, fail to segment parts correctly, or are confined to overly simplistic objects (Kerr et al., 2024; Song et al., 2024; Peng et al., 2025), leaving robust, general-purpose reconstruction from monocular video a largely unsolved frontier. To break this ambiguity, motion priors from tracking models offer a promising direction. Previous methods, such as Shape-of-Motion (Wang et al., 2024a) and ArtiPoint (Werby et al., 2025), have explored lifting 2D tracks for supervision. More recently, the advent of powerful perception models like SpatialTrackerV2 (Xiao et al., 2025) and TAPIP3D (Zhang et al., 2025) provides 3D tracks, which offer richer motion information. However, both lifted tracks and 3D tracks contain substantial noise that makes them ineffective for direct use in articulated object reconstruction, leaving the problem of how to effectively leverage them as motion priors unexplored.

To address these challenges, we propose VideoArtGS, which introduces several key innovations for reconstructing articulated objects from monocular video. Central to our approach are two key insights: (1) motion priors from pre-trained tracking models are essential for disambiguating object movement, and (2) by enforcing articulation constraints (e.g., linear or circular trajectories for prismatic and revolute joints), we leverage both object-part movement priors and the reconstruction objective to jointly suppress noise in the tracks, recover structural cues of the moving parts. Specifically, we design a novel motion prior guidance pipeline that analyzes raw 3D tracking trajectories, filters noise, classifies motion types (e.g., revolute, prismatic), and clusters points into coherent parts. This process yields accurate initial estimates for the joint parameters and part centers, transforming the intractable joint optimization into a well-posed refinement problem. To further enhance reconstruction quality, we design a hybrid center-grid part assignment module. This module combines the strengths of spatial clustering for distinct movable parts with a flexible grid-based representation to model complex 3D geometry of objects, enabling clean part segmentation and precise deformation modeling.

These designs enable VideoArtGS to achieve state-of-the-art performance, reducing reconstruction and articulation estimation errors by approximately two orders of magnitude compared to previous methods on both simple two-part objects and on our new, challenging VideoArtGS-20 dataset. Our approach opens new possibilities for practical digital twin creation from readily available video data, with applications in scenarios where multi-state capture is impractical or impossible. Through extensive experiments, we demonstrate the effectiveness of our method in delivering high-quality reconstruction of articulated objects from monocular video sequences.

**Contributions**   Our main contributions of this work can be summarized as follows:

- We propose VideoArtGS, a novel method for articulated object reconstruction from monocular video that achieves state-of-the-art performance, reducing key error metrics by up to two orders of magnitude over prior work.
- We introduce a motion prior guidance framework that analyzes 3D tracking trajectories to robustly initialize the deformation field, making the ill-posed reconstruction problem tractable. We design a hybrid center-grid part assignment module that accurately segments parts and benefit articulation learning, accommodating complex geometries.
- We conduct extensive experiments and establish a new benchmark for video-based articulated object reconstruction, validating the practical applicability of our approach. Our comprehensive ablation studies systematically validate our designs and point out directions for future improvement.

## 2 RELATED WORK

### 2.1 DYNAMIC SCENE RECONSTRUCTION

Dynamic scene reconstruction is a long-standing challenge in computer vision. A significant line of work focuses on jointly estimating camera poses and scene geometry, often represented as depth maps

or point clouds. Pioneering methods like DROID-SLAM (Teed & Deng, 2021), CasualSAM (Tang et al., 2025), and Mega-SaM (Li et al., 2025) established robust frameworks for this task. More recently, foundation models have emerged, with DUSt3R (Wang et al., 2024b) and VGGT (Wang et al., 2025b) providing a powerful basis for 3D reconstruction. Subsequent works like MonST3R (Zhang et al., 2024), CUT3R (Wang et al., 2025d), and SpatialTrackerV2 (Xiao et al., 2025) have fine-tuned or extended DUSt3R or VGGT to better handle dynamic content.

While the above methods provide camera and geometry information, representing the dynamic scene itself has been revolutionized by 3D Gaussian Splatting (Kerbl et al., 2023). Many 4D extensions learn to deform Gaussians implicitly over time (Jung et al., 2023; Katsumata et al., 2023; Wu et al., 2024; Luiten et al., 2024; Li et al., 2024; Lu et al., 2024; Lei et al., 2024a; Guo et al., 2024; Qian et al., 2024; Bae et al., 2024; LIU et al., 2025; Wu et al., 2025), which excels at capturing complex, non-rigid motion but offers no explicit control over an object's underlying structure. Although some methods learn dense tracks by reconstructing videos (Wang et al., 2025c; Lei et al., 2024b), they do not model the articulated object and cannot reconstruct interactive assets from it. Attempts to add control via superpoints (Huang et al., 2024) or physics engines (Xie et al., 2024; Jiang et al., 2024) have been made, but they either fail to extract accurate physical parameters or require impractical priors. VideoArtGS bridges this gap by integrating an explicit articulation model directly into the deformable Gaussian framework, enabling high-fidelity reconstruction for articulated objects.

## 2.2 ARTICULATED OBJECT RECONSTRUCTION

Reconstructing articulated objects presents a dual challenge: one must solve for both the part-level geometry and the underlying articulation parameters. One family of methods employs end-to-end models to predict both part segmentation and joint parameters (Heppert et al., 2023; Wei et al., 2022; Kawana et al., 2021; Mandi et al., 2024; Jiang et al., 2022; Ma et al., 2023; Nie et al., 2022; Hsu et al., 2023; Goyal et al., 2025; Xia et al., 2024), while some similar methods only predict articulation parameters (Hu et al., 2017; Yi et al., 2018; Li et al., 2020; Wang et al., 2019; Sun et al., 2023; Liu et al., 2022; Weng et al., 2021; Sturm et al., 2011; Chu et al., 2023; Martín-Martín et al., 2016; Liu et al., 2023c; Gadre et al., 2021; Mo et al., 2021; Jain et al., 2021; Yan et al., 2020; Lei et al., 2023). Their fundamental limitation, however, is a reliance on large, annotated datasets, which prevents them from generalizing to unseen object categories. The dominant paradigm relies on multi-view observations at discrete multi-state (Liu et al., 2025; Tseng et al., 2022; Mu et al., 2021; Lewis et al., 2022; Liu et al., 2023a; Lei et al., 2024a; Deng et al., 2024; Swaminathan et al., 2024; Noguchi et al., 2022; Zhang et al., 2021; Pillai et al., 2015; Liu et al., 2023b; Wang et al., 2025a; Lewis et al., 2025; Zhang & Lee, 2025). These methods leverage strong geometric constraints, which simplify the problem but require impractical and controlled data capture setups. A more practical but far more challenging setting is reconstruction from a monocular video. Existing video-based methods are typically limited to simple objects (Song et al., 2024; Peng et al., 2025) or rely on a pre-trained segmentation model that has limited generalization ability (Zhou et al., 2025). In contrast, VideoArtGS is designed for this challenging setting. By introducing a robust motion prior guidance pipeline, we effectively disentangle the scene dynamics and transform the ill-posed problem into a tractable one, achieving state-of-the-art results where prior methods have struggled.

## 3 METHOD

Given a monocular video sequence $\{I_t\}_{t=1}^{T}$, VideoArtGS reconstructs articulated objects with part meshes $\mathcal{M}$ and articulation parameters $\Psi$. We first use the VGGT (Wang et al., 2025b) trained for dynamic scenes from SpatialTrackerV2 (Xiao et al., 2025) to estimate the depths and camera poses, and then reconstruct the object with 3D Gaussians $\mathcal{G} = \{G_i\}_{i=1}^{N}$ and an articulation-based deformation field $\mathcal{F}$. This field contains a part segmentation module $S_\phi$ and articulation parameters $\Psi$ (including axis directions $\boldsymbol{d}$, axis origins $\boldsymbol{o}$, and time-variant joint states $\theta_t$) that control the dynamics of each part. We also introduce motion prior from a pre-trained tracking model TAPIP3D (Zhang et al., 2025) to guide the initialization and optimization of the deformation field. An overview of VideoArtGS is presented in Fig. 1, with details on key designs provided in the following sections.

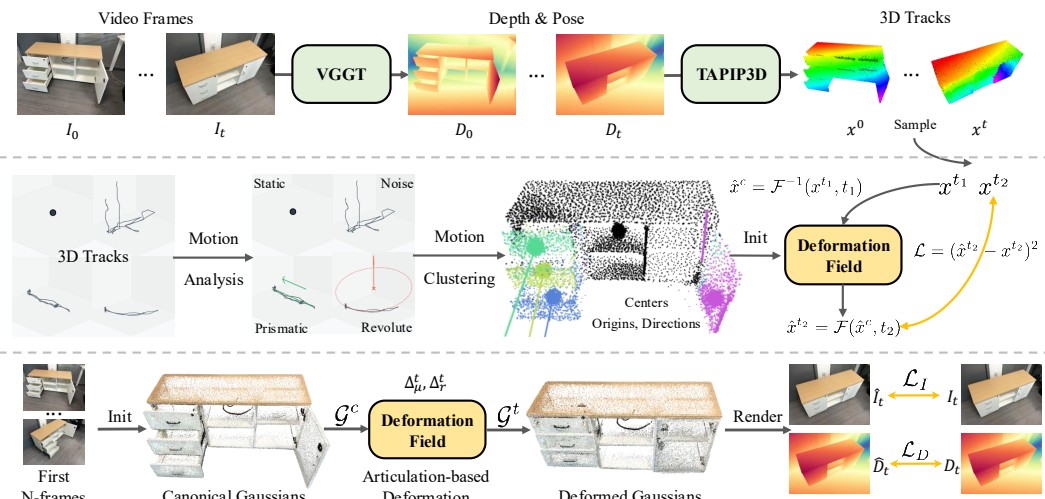

Figure 1: **The overview of VideoArtGS.** Given video frames, we first use VGGT (Wang et al., 2025b) to estimate the depths along with camera poses and then use TAPIP3D (Zhang et al., 2025) to obtain 3D tracks. We design a motion prior guidance pipeline to analyze and group these tracks, initializing our articulation-based deformation field with motion information and optimizing it with tracking loss. Finally, we reconstruct the object with 3D Gaussians and the deformation field, jointly optimizing both modules by rendering and tracking loss.

## 3.1 ARTICULATION-BASED DEFORMATION FIELD

To model the temporal dynamics of an articulated object, we formulate an articulation-based deformation field $\mathcal{F}$ that transforms a set of canonical Gaussians $G_i^c = \{\boldsymbol{\mu}_i^c, \boldsymbol{r}_i^c, \boldsymbol{s}_i, \sigma_i, \boldsymbol{h}_i\}$ into the observation state $G_i^t = \{\boldsymbol{\mu}_i^t, \boldsymbol{r}_i^t, \boldsymbol{s}_i, \sigma_i, \boldsymbol{h}_i\}$ for any given time $t$. Since articulation is a rigid process, the intrinsic properties of each Gaussian—its scale ($\boldsymbol{s}_i^c$), opacity ($\sigma_i^c$), and appearance ($\boldsymbol{h}_i^c$)—are treated as time-invariant, while its position ($\boldsymbol{\mu}_i^t$) and rotation ($\boldsymbol{r}_i^t$) are time-variant. Following ArtGS (Liu et al., 2025), VideoArtGS first assigns each Gaussian to object parts through a segmentation module $S_\phi(\cdot)$ and then applies the corresponding rigid transformation for each part:

$$\boldsymbol{m}_i = S_\phi(G_i^c), \quad G_i^t = \sum_{k=1}^{K} m_{ik} \cdot \mathcal{T}_k^t(G_i^c) \tag{1}$$

where $\boldsymbol{m}_i = [m_{i1}, \dots, m_{iK}]$ represents the assignment probabilities of $i$-th Gaussian to $K$ parts, and $\mathcal{T}_k^t$ denotes the rigid transformation for $k$-th part at time $t$. The number of movable parts and joint types (revolute or prismatic) could be obtained by GPT4-o (Hurst et al., 2024). We provide more details of articulation modeling in Appendix A.3.

**Hybrid Center-grid Part Assignment** To effectively assign Gaussians to articulable parts, ArtGS (Liu et al., 2025) proposes a center-based part assignment module that segments parts using the Mahalanobis distance between Gaussians and learnable centers. However, this approach faces limitations when the static base part has complex geometries. A simple but key observation is that static regions remain fixed in space. Unlike movable parts that naturally form distinct motion clusters, the static base part is better characterized by its fixed spatial volume rather than a movable center. We therefore propose a hybrid center-grid part assignment module that combines two strategies. For the $K-1$ movable parts, we define learnable part centers $C_k = (\boldsymbol{p}_k, \boldsymbol{V}_k, \boldsymbol{\lambda}_k)$ with center location $\boldsymbol{p}_k \in \mathbb{R}^3$, rotation matrix $\boldsymbol{V}_k \in \mathbb{R}^{3\times3}$, and scale vector $\boldsymbol{\lambda}_k \in \mathbb{R}^3$. For the static base part, we use a learnable hash grid $H$ to model its spatial region directly. Given the canonical position $\boldsymbol{\mu}_i^c$ of a Gaussian $G_i^c$, we compute its assignment probabilities $\boldsymbol{m}_i$ by fusing scores from both models. First, we compute the squared Mahalanobis distance $\boldsymbol{D}_{i,k}$ to each of the $K-1$ movable part centers:

$$\boldsymbol{D}_{i,k} = \left(\frac{\boldsymbol{V}_k(\boldsymbol{\mu}_i^c - \boldsymbol{p}_k)}{\boldsymbol{\lambda}_k}\right)^\top \left(\frac{\boldsymbol{V}_k(\boldsymbol{\mu}_i^c - \boldsymbol{p}_k)}{\boldsymbol{\lambda}k}\right) + \Delta_{i,k}, \tag{2}$$

where $\Delta_{i,k}$ is a residual term for improving boundary identification that is introduced by ArtGS (Liu et al., 2025). Simultaneously, we query the hash grid at the Gaussian's position to get a feature vector,

which is processed by a small MLP to produce a single logit, $l_i = \text{MLP}(H(\boldsymbol{\mu}_i^c))$, representing the "staticness" score. The final assignment probabilities $\boldsymbol{m}_i \in \mathbb{R}^K$ are obtained by concatenating the static logit with the negative distances of the movable parts and applying a softmax function:

$$\boldsymbol{m}_i = \text{Softmax}\left(\text{concat}\left([l_i, -\boldsymbol{D}_{i,1}, \ldots, -\boldsymbol{D}_{i,K-1}]\right)\right). \tag{3}$$

This hybrid formulation enables robust segmentation by leveraging both structured geometric relationships for movable parts and flexible spatial modeling for complex static regions.

### 3.2 Motion Prior Guidance

We use a pre-trained tracking model TAPIP3D (Zhang et al., 2025) to obtain 3D tracking trajectories, providing a motion prior to guide the initialization and optimization of the deformation field.

**Motion Pattern Analysis.** To identify noises and extract motion information from tracking trajectories, we first analyze the motion pattern of each trajectory and divide all trajectories into 4 classes: static, prismatic, revolute, and noise. If the maximum displacement distance of the $i$-th trajectory $\{\boldsymbol{x}_i^t\}_{t=1}^T$ is below a threshold $\epsilon_s$, it is classified as a static trajectory. For the remaining dynamic trajectories, we use line fitting and circle fitting to identify the motion type and motion parameters. A main challenge is that all points remain static for most of the time and move for only a short period of time, which is particularly prominent for objects with multiple parts. Many points are concentrated in the same area, leading to the fitting collapse. To deal with this problem, we design an adaptive spatial downsampling approach. Specifically, we first voxelize each trajectory, and then retain only one point in each voxel. To handle different ranges of trajectories, we dynamically adjust the voxel size based on the range of the trajectory. After downsampling, we use the remaining points to fit the trajectory.

For prismatic motion, we use Principal Component Analysis (PCA) for line fitting, combined with the RANSAC algorithm to improve robustness. For revolute motion, we first fit the best plane, then fit a circle on that plane. We use Singular Value Decomposition (SVD) to find the normal vector and verify whether the trajectory conforms to rigid rotation. If the line/circle fitting error of a trajectory is less than pre-defined thresholds $\epsilon_l/\epsilon_c$, it is considered a valid track; otherwise, it is treated as a noise track. For a valid track, we prioritize models with smaller fitting errors. The above process also provides the direction of prismatic trajectories and the direction and origin of revolute trajectories.

**Motion Clustering.** Given valid trajectories with their motion type and motion parameters, we construct feature vectors and then use K-means clustering to group trajectories into different parts. For prismatic motion, the feature vector contains starting position, average position, motion direction, and normalized velocity. For revolute motion, the feature vector contains starting position, average position, axis direction, axis origin, and angular velocity. To improve clustering quality, we adopt an iterative filtering strategy, combining directional angle filtering and Euclidean distance filtering to remove outliers. Finally, we generate articulation information for core parameters initialization of the deformation field $\mathcal{F}$, including the axis direction $\boldsymbol{d}$, axis origin $\boldsymbol{o}$, and part centers $\boldsymbol{p}$.

**Deformation Field Initialization.** We randomly initialize the remaining parameters of $\mathcal{F}$ and then use the tracking trajectories to optimize them. We design two different losses $\mathcal{L}_{c2o}$ and $\mathcal{L}_{o2o}$ to optimize the deformation field. $\mathcal{L}_{c2o}$ is the canonical-to-observation loss, which provides direct supervision for the deformation from canonical state to the observation state:

$$\hat{\boldsymbol{x}}_i^t = \mathcal{F}(\boldsymbol{x}_i^c, t), \quad \mathcal{L}_{c2o} = \frac{1}{N}\sum_{i=1}^N (\boldsymbol{x}_i^t - \hat{\boldsymbol{x}}_i^t)^2, \tag{4}$$

where $\boldsymbol{x}_i^c, \boldsymbol{x}_i^t$ are point positions sampled from trajectory $\{\boldsymbol{x}_i^t\}_{t=1}^T$. $\mathcal{L}_{o2o}$ is the observation-to-observation loss, which enhances temporal consistency between two observation states $t_0$ and $t_1$:

$$\hat{\boldsymbol{x}}_i^c = \mathcal{F}^{-1}(\boldsymbol{x}_i^{t_0}, t_0), \quad \hat{\boldsymbol{x}}_i^{t_1} = \mathcal{F}(\hat{\boldsymbol{x}}_i^c, t_1), \quad \mathcal{L}_{o2o} = \frac{1}{N}\sum_{i=1}^N (\boldsymbol{x}_i^{t_1} - \hat{\boldsymbol{x}}_i^{t_1})^2, \tag{5}$$

where $\mathcal{F}^{-1}$ is the inversed deformation field of $\mathcal{F}$ and $\mathcal{F}^{-1}$ shares the same parameters with $\mathcal{F}$. See Appendix A.4 for details of the inverse deformation field. We randomly sample pairs of tracking trajectories at time $(c, t)$ for $\mathcal{L}_{c2o}$ and $(t_0, t_1)$ within a 30-frame window for $\mathcal{L}_{o2o}$ to robustly optimize $\mathcal{F}$. The final tracking loss could be calculated by:

$$\mathcal{L}_{\text{track}} = \mathcal{L}_{c2o} + \mathcal{L}_{o2o}. \tag{6}$$

### 3.3 OPTIMIZATION

To reconstruct high-quality geometry of objects, we assume the object remains static in the first $N$ frames and initialize canonical Gaussians $\mathcal{G}^c$ with these frames. We train $\mathcal{G}^c$ with the rendering loss $\mathcal{L}_{\text{render}} = (1 - \lambda_{\text{SSIM}})\mathcal{L}_1 + \lambda_{\text{SSIM}}\mathcal{L}_{\text{D-SSIM}} + \mathcal{L}_D$ used in ArtGS (Liu et al., 2025), where $\mathcal{L}_D = \log\left(1 + ||\boldsymbol{D} - \bar{\boldsymbol{D}}||_1\right)$ is a depth loss. After initializing the deformation field $\mathcal{F}$ and canonical Gaussians $\mathcal{G}^c$, we jointly optimize $\mathcal{F}$ and $\mathcal{G}^c$ across all video frames and tracking trajectories with rendering loss $\mathcal{L}_{\text{render}}$ and canonical-to-observation tracking loss $\mathcal{L}_{c2o}$:

$$\mathcal{L} = \mathcal{L}_{\text{render}} + \lambda_{c2o}\mathcal{L}_{c2o}. \tag{7}$$

We provide more implementation and model training details in Appendix A.

## 4 EXPERIMENTS

**Datasets**  We conduct a comprehensive evaluation on two distinct datasets to assess the performance of existing methods on objects with varying articulation complexity. (1) Video2Articulation-S, a dataset proposed by (Peng et al., 2025), which serves as our benchmark for simple articulated objects. It consists of 73 test videos across 11 categories of synthetic objects from the PartNet-Mobility dataset (Xiang et al., 2020), where each object has only a single movable part. (2) VideoArtGS-20, a newly curated dataset that evaluates the performance on more complex scenarios. It contains 20 videos of complex articulated objects of 10 categories from PartNet-Mobility, featuring more challenging kinematics with 2 to 9 movable parts per object.

**Metrics**  Our evaluation protocol includes metrics for both articulation estimation and mesh reconstruction quality. For articulation estimation, we measure axis direction error (deg), axis position error (cm), and joint state error (deg for revolute joints, cm for prismatic joints) between the predicted and ground-truth joint parameters. For mesh reconstruction, we assess geometric accuracy using the bi-directional Chamfer Distance (CD). This is computed between the reconstructed mesh and the ground-truth mesh, using 10,000 points uniformly sampled from each surface. We report the CD (in cm) for the whole object (CD-w), the static part (CD-s), and the movable parts (CD-m).

### 4.1 RESULTS ON SIMPLE ARTICULATED OBJECTS

**Experimental Setup**  For this benchmark, we use the Video2Articulation-S dataset. We perform a quantitative comparison against three state-of-the-art methods: ArticulateAnything (Le et al., 2025), RSRD (Kerr et al., 2024), and Video2Articulation (Peng et al., 2025). Following the standard evaluation protocol established by Video2Articulation (Peng et al., 2025), all metrics are reported as the mean and standard deviation (mean ± std) across all test videos, and articulation estimation metrics are divided into revolute and prismatic. For a fair and direct comparison, our experimental setup utilizes ground-truth depth and camera poses, and the results for all baseline methods are taken directly from Video2Articulation (Peng et al., 2025). To ensure consistency with our evaluation, we have converted their reported metrics from meters (m) to centimeters (cm) and from radians to degrees. We also retrain and evaluate VideoAticulation on this dataset.

**Results and Analysis**  The quantitative results, presented in Tab. 1, demonstrate that our method substantially outperforms all baseline approaches across all metrics. The most significant gains are in joint parameter estimation, where VideoArtGS achieves an order-of-magnitude reduction in error compared to the second-best method, Video2Articulation. This dramatic increase in accuracy is primarily attributable to our motion prior guidance, which provides an accurate starting point for optimization that prior methods lack. Our method also achieves a new state of the art in reconstruction quality. The exceptional improvements on both movable parts and the static part validate the effectiveness of VideoArtGS. As illustrated in Fig. 2, our method consistently produces high-fidelity mesh reconstructions with clean part boundaries and precise articulation. This demonstrates the robustness and high quality of our approach across the diverse object categories.

### 4.2 RESULTS ON COMPLEX ARTICULATED OBJECTS

**Experimental Setup**  We conduct an evaluation on our newly curated VideoArtGS-20 dataset, which contains complex, multi-part objects. We compare our method against current state-of-the-

Table 1: **Quantitative evaluation on Video2Articulation-S dataset.** Metrics are reported as mean $\pm$ std over all test videos. Lower ($\downarrow$) is better on all metrics, and the **best results** are highlighted in bold. [†] means the results are taken from VideoAticulation (Peng et al., 2025).

| Method | Revolute Joint Estimation | | | Prismatic Joint Estimation | | Reconstruction | | |
|---|---|---|---|---|---|---|---|---|
| | Axis (∘) | Position (cm) | State (∘) | Axis (deg) | State (cm) | CD-w (cm) | CD-m (cm) | CD-s (cm) |
| ArticulateAnything[†] (Le et al., 2025) | 46.98±45.27 | 81.00±40.00 | N/A | 52.71±44.69 | N/A | 11.00±22.00 | 59.00±73.00 | 7.00±18.00 |
| RSRD[†] (Kerr et al., 2024) | 67.06±29.22 | 203.00±748.00 | 59.02±34.38 | 69.91±24.07 | 70.00±48.00 | 339.00±2147.00 | 82.00±117.00 | 14.00±41.00 |
| Video2Articulation[†] (Peng et al., 2025) | 18.34±32.09 | 13.00±25.00 | 14.32±26.35 | 13.75±18.91 | 8.00±22.00 | 1.00±1.00 | 13.00±26.00 | 6.00±19.00 |
| Video2Articulation (Peng et al., 2025) | 13.83±28.15 | 11.55±22.39 | 10.25±21.27 | 14.37±19.08 | 3.44±6.25 | 3.45±16.46 | 12.21±24.44 | 5.39±17.09 |
| **Ours** | **0.32±0.44** | **0.42±0.75** | **1.15±2.29** | **0.35±0.45** | **1.03±2.46** | **0.29±0.24** | **0.40±0.32** | **1.11±2.11** |

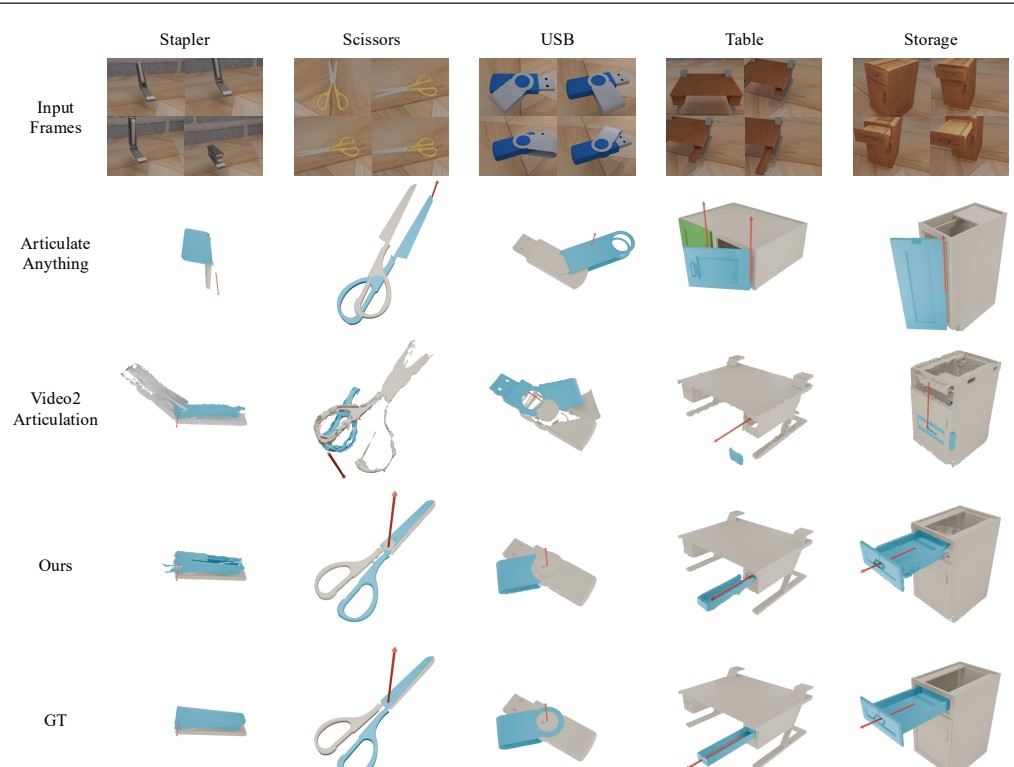

Figure 2: **Qualitative results on Video2Articulation-S dataset.** We present reconstruction comparisons between baselines and our model on the Video2Articulation-S dataset.

art methods ArticulateAnything (Le et al., 2025) and Video2Articulation (Peng et al., 2025). As RSRD (Kerr et al., 2024) failed to correctly segment parts, we don't use it as a baseline. All metrics are averaged across all parts and reported as mean ± std over all objects. A critical limitation of prior work is that Video2Articulation (Peng et al., 2025) is designed only for a single movable part. To establish a baseline, we extend it to multi-part objects: we manually isolate video segments where only a single part is in motion and then merge the moving map to extract multiple part meshes.

**Results and Analysis** On the complex, multi-part VideoArtGS-20 dataset, our method's advantages become even more pronounced. Compared to Video2Articulation-S, VideoArtGS-20 has larger camera motion and includes more moving parts, posing a greater challenge to existing baselines. As shown in Fig. 3, Video2Articulation struggles to accurately segment moving parts, while ArticulateAnything often retrieves incorrect parts. As demonstrated in Tab. 2, VideoArtGS achieves state-of-the-art performance, drastically outperforming baselines in this complex multi-part setting. It is critical to note that the retrieval database of ArticulateAnything (Le et al., 2025) contains the ground-truth meshes and joints from PartNet-Mobility, the same source as our test data. Despite this near-oracle condition for the baseline, our method still reduces articulation estimation errors by nearly two orders of magnitude and excels at mesh reconstruction where baselines fail. These results confirm that VideoArtGS's advantages generalize from simple to complex scenarios, providing a robust and scalable solution for reconstructing articulated objects from monocular video.

Table 2: **Quantitative evaluation on VideoArtGS-20 dataset.** Metrics are reported as mean $\pm$ std over all test videos. Lower ($\downarrow$) is better on all metrics, and the **best results** are highlighted in bold.

| Method | Axis (°) | Position(cm) | CD-w (cm) | CD-m (cm) | CD-s (cm) |
|---|---|---|---|---|---|
| ArticulateAnything (Le et al., 2025) | 43.65 ± 44.72 | 15.66 ± 36.20 | 16.10 ± 37.34 | 17.66 ± 36.74 | 16.04 ± 37.36 |
| Video2Articulation (Peng et al., 2025) | 48.88 ± 24.18 | 37.04 ± 31.82 | 5.07 ± 21.78 | 30.63 ± 25.64 | 10.22 ± 22.23 |
| **Ours** | **0.34±0.80** | **0.10±0.10** | **0.09±0.09** | **0.26±0.61** | **0.24±0.58** |

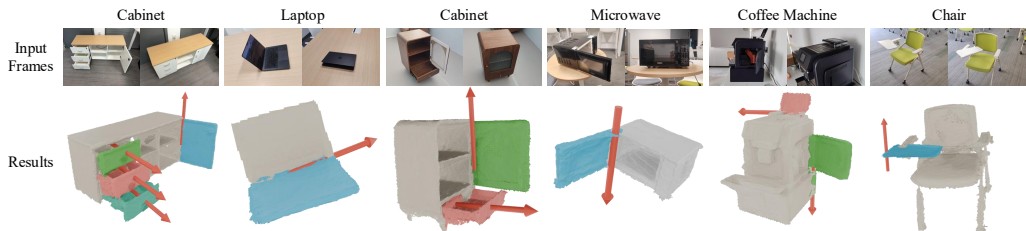

Figure 3: **Qualitative results on VideoArtGS-20 dataset.** We present reconstruction comparisons between baselines and our model on the VideoArtGS-20 dataset.

Figure 4: **Qualitative results on real-world data.** We present reconstruction results of our model on real-world data, including both simple two-part and complex multi-part objects.

## 4.3 RESULTS ON REAL-WORLD DATA

**Experimental Setup** We also validate the effectiveness of VideoArtGS on real-world data. We capture monocular videos using a mobile phone camera without LiDAR. We use articulated objects of different categories with different numbers of joints to verify the generalization ability of our method. The input to our model is solely the monocular RGB video.

**Results and Analysis** As shown in Fig. 4, our VideoArtGS successfully reconstructs a diverse set of articulated objects from self-captured, real-world monocular videos, building digital twins with high-fidelity geometry and accurate articulation parameters. VideoArtGS effectively decouples the object's geometry from its time-varying motion, enabling the creation of a controllable digital asset, fulfilling the promise of creating truly interactable digital twins from casual video captures.

## 4.4 ABLATION STUDIES

**Experimental Setup** To validate the effectiveness of each component in our method, we conduct comprehensive ablation studies on the VideoArtGS-20 dataset. We systematically remove different components and analyze their impact on performance, with all metrics reported as mean ± std.

Table 3: **Ablation studies on VideoArtGS-20 dataset.** Lower (↓) is better on all metrics, and the **best results** are highlighted in bold.

| Method | Axis (°) | Position (cm) | CD-w (cm) | CD-m (cm) | CD-s (cm) |
|---|---|---|---|---|---|
| Ours | **0.34±0.80** | **0.10±0.10** | **0.09±0.09** | 0.26±0.61 | **0.24±0.58** |
| w/o motion prior | 55.28±15.49 | 23.74±17.49 | 10.18±29.94 | 87.77±17.02 | 14.37±29.69 |
| w/o center init | 20.64±21.64 | 22.42±25.03 | 10.33±29.90 | 83.32±14.06 | 14.07±29.80 |
| w/o deform init | 3.96±3.73 | 2.45±3.07 | 0.11±0.12 | 1.50±2.71 | 0.72±2.05 |
| w/o axis init | 0.60±1.26 | 0.86±2.58 | **0.09±0.09** | **0.25±0.56** | 0.27±0.74 |
| w/o hybrid | 1.21±1.77 | 2.51±10.47 | 0.15±0.29 | 10.35±23.84 | 0.50±1.08 |
| w/o $\mathcal{L}_{o2o}$ | 0.68±1.55 | 0.57±1.85 | 0.11±0.12 | 0.58±1.03 | 0.27±0.65 |
| w/o $\mathcal{L}_{c2o}$ | 0.40±0.79 | 0.13±0.11 | 0.09±0.10 | 0.35±0.86 | 0.26±0.70 |

**Results and Analysis** The results, summarized in Tab. 3, systematically deconstruct our model's performance and validate the critical role of our core design choices.

- *Motion Prior Guidance.* The most profound impact comes from removing the entire motion prior guidance (w/o motion prior), including the initialization of centers, joint axes, and deformation field, which results in a catastrophic failure of the model. This unequivocally confirms our central hypothesis: without a strong initial estimate derived from motion cues, the optimization problem of complex articulated objects is intractable.
- *Initialization of components.* Removing the part centers initialization (w/o center init) leads to a complete failure, underscoring the necessity of establishing a correct spatial anchor for each part before optimizing its motion. Removing the deformation field initialization (w/o deform init) causes a notable but not catastrophic performance drop, particularly on movable part reconstruction. Interestingly, removing the axis initialization (w/o axis init) yields a marginal drop in articulation estimation and has a minimal effect on reconstruction. This suggests that the framework is robust enough to find the correct axis if the part centers and correspondences are well-initialized, though direct initialization remains beneficial for stability and performance.
- *Hybrid Center-Grid Assignment.* Replacing the hybrid center-grid assignment module with the center-based assignment module (w/o hybrid) leads to moderate performance drops, especially for the reconstruction of movable parts and articulation estimation. This result highlights that our hybrid assignment is essential for correctly segmenting parts and learning articulation dynamics.
- *Tracking Losses.* Disabling the observation-to-observation tracking loss (w/o $\mathcal{L}_{o2o}$) degrades performance more than disabling the direct canonical-to-observation loss (w/o $\mathcal{L}_{c2o}$). This indicates that enforcing temporal consistency directly on the observation space is a more critical constraint for achieving precise and stable joint estimation.

These ablation results confirm that our method's success relies on the synergistic combination of all components. The results unequivocally demonstrate that the motion prior guidance and the hybrid part assignment are the two foundational pillars enabling our method's success. The remaining components, while having a smaller individual impact, contribute synergistically to the stability and precision of the final result, solidifying the robustness of our overall framework.

## 5 CONCLUSION

In conclusion, we introduce VideoArtGS, a novel method that reconstructs high-fidelity articulated objects from a monocular video. We solve the fundamentally ill-posed challenge by introducing a motion prior guidance pipeline, leveraging 3D tracks to provide robust initialization and optimization of the deformation field. Combined with a hybrid center-grid assignment module for accurate part segmentation, VideoArtGS achieves a new state of the art, reducing key error metrics by up to two orders of magnitude and validating on our new, challenging VideoArtGS-20 benchmark. While VideoArtGS sets a new performance benchmark, its reliance on upstream trackers, pose estimators, and the necessity of visible motion in the video present avenues for future work. Promising directions include developing end-to-end models that jointly learn tracking and reconstruction or integrating physical priors to handle more challenging, motion-scarce scenarios.

**Reproducibility Statement**  We re-executed all experiments before submission to ensure reproducibility and consistency of our results. Detailed implementation and training procedures are provided in the appendix. Upon paper acceptance, we will release all code, data, and model weights publicly. The planned code release will include training scripts, evaluation protocols, and detailed documentation to facilitate easy reproduction of all experimental results.

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

## A IMPLEMENTATION AND TRAINING DETAILS

### A.1 VIDEOARTGS-20 DATASET

We introduce and evaluate our method on VideoArtGS-20, a newly curated dataset featuring 10 object categories: Faucet, Door, Refrigerator, Table, Storage Furniture, Bucket, Eyeglasses, Oven, Window, and Printer. For each object, we render a monocular video with 150 static frames in different viewpoints and 60 dynamic frames for each movable part. The dataset provides a challenging benchmark with objects containing up to 10 parts and 9 movable joints. Further details and visualizations are available in Tab. A.1 and Fig. A.1.

### A.2 3D GAUSSIAN SPLATTING

3D Gaussian Splatting (3DGS) represents a 3D scene using a collection of 3D Gaussians (Kerbl et al., 2023). Each Gaussian $G_i$ is parameterized by its center $\boldsymbol{\mu}_i \in \mathbb{R}^3$, covariance matrix $\boldsymbol{\Sigma}_i \in \mathbb{R}^{3\times3}$, opacity $\sigma_i \in [0,1]$, and spherical harmonics coefficients $\boldsymbol{h}_i$ for view-dependent color. The opacity of a 3D Gaussian at spatial point $\boldsymbol{x}$ is computed as:

$$\alpha_i(\boldsymbol{x}) = \sigma_i \exp\left(-\frac{1}{2}(\boldsymbol{x}-\boldsymbol{\mu}_i)^\top \boldsymbol{\Sigma}_i^{-1}(\boldsymbol{x}-\boldsymbol{\mu}_i)\right), \quad \text{where} \quad \boldsymbol{\Sigma}_i = \boldsymbol{R}_i \boldsymbol{S}_i \boldsymbol{S}_i^\top \boldsymbol{R}_i^\top. \tag{A.1}$$

To ensure $\boldsymbol{\Sigma}_i$ remains positive semi-definite, it is decomposed into a rotation matrix $\boldsymbol{R}_i$ (parameterized by quaternion $\boldsymbol{r}_i$) and a scaling diagonal matrix $\boldsymbol{S}_i$ (parameterized by scale vector $\boldsymbol{s}_i$). To render an image, 3D Gaussians are projected onto the 2D image plane and aggregated using $\alpha$-blending:

$$\boldsymbol{I} = \sum_{i=1}^{N} T_i \alpha_i^{\text{2D}} \mathcal{SH}(\boldsymbol{h}_i, \boldsymbol{v}_i), \quad \text{where} \quad T_i = \prod_{j=1}^{i-1}(1-\alpha_j^{\text{2D}}). \tag{A.2}$$

Here, $\alpha_i^{\text{2D}}$ is the 2D version of Eq. (A.1), $\mathcal{SH}(\cdot)$ calculates spherical harmonics for view direction $\boldsymbol{v}_i$. Given multi-view images $\{\bar{\boldsymbol{I}}_i\}_{i=1}^{N}$, 3DGS optimizes the paramters using L1 loss and D-SSIM loss (Kerbl et al., 2023) with a loss weight $\lambda_{\text{SSIM}}$:

$$\mathcal{L}_I = (1-\lambda_{\text{SSIM}})\mathcal{L}_1 + \lambda_{\text{SSIM}}\mathcal{L}_{\text{D-SSIM}}, \tag{A.3}$$

### A.3 ARTICULATION MODELING

**Articulation Modeling**    Building upon the part assignments, we model articulation through learnable joint parameters, including axis direction $\boldsymbol{d}$, axis origin $\boldsymbol{o}$, and time-variant joint state $\theta^t$. To learn a smooth trajectory of joint states, we model it with Foriour embedding $E(\cdot)$ followed by a learnable MLP: $\theta^t = \text{MLP}(E(t))$. We represent the rigid transformation as dual-quarternion $\boldsymbol{q}^t = (\boldsymbol{q}_r^t, \boldsymbol{q}_d^t)$ for smooth skinning, where $\boldsymbol{q}_r^t, \boldsymbol{q}_p^t$ represent the rotation and translation components respectively. The dual-quaternion of each joint could be calculated as:

$$\begin{aligned} \text{prismatic}: \boldsymbol{q}_r^t &= (1,0,0,0), \quad \bar{\boldsymbol{o}}^t = (0,\theta^t \cdot \boldsymbol{d}), \quad \boldsymbol{q}_d^t = 0.5 \cdot \bar{\boldsymbol{o}}^t \otimes \boldsymbol{q}_r^t, \\ \text{revolute}: \boldsymbol{q}_r^t &= (\cos\frac{\theta^t}{2}, \sin\frac{\theta^t}{2}\cdot\boldsymbol{d}), \quad \bar{\boldsymbol{o}}^t = (0,\boldsymbol{o}), \quad \boldsymbol{q}_d^t = 0.5\cdot(\bar{\boldsymbol{o}}^t\otimes\boldsymbol{q}_r^t - \boldsymbol{q}_r^t\otimes\bar{\boldsymbol{o}}^t). \end{aligned} \tag{A.4}$$

Then we calculate the per-gaussian dual-quaternion $\boldsymbol{q}_i^t$ with part assignment probabilities $\boldsymbol{m}_i$ by:

$$\boldsymbol{q}_i^t = \sum_{k=1}^{K} m_{ik} \cdot \boldsymbol{q}_k^t = \left(\sum_{k=1}^{K} m_{ik}\cdot\boldsymbol{q}_{k,r}^t, \sum_{k=1}^{K} m_{ik}\cdot\boldsymbol{q}_{k,d}^t\right). \tag{A.5}$$

where $\boldsymbol{q}_k^t$ is the dual-quarternion of $k$-th part. The position and rotation of Gaussian $G_i^t$ are obtained by:

$$\boldsymbol{\mu}_i^t = \boldsymbol{R}_i^t \cdot \boldsymbol{\mu}_i^c + \boldsymbol{t}_i^t, \quad \boldsymbol{r}_i^t = \boldsymbol{q}_{i,r}^t \otimes \boldsymbol{r}_i^c, \tag{A.6}$$

where $\boldsymbol{R}_i^t$ and $\boldsymbol{t}_i^t$ is rotation matrix and translation vector derived from $\boldsymbol{q}_i^t$, and $\otimes$ denotes quaternion multiplication operation. We provide the detailed derivation process of dual-quaternion in the following paragraphs.

Table A.1: Dataset configuration.

| Object ID | Category | #Part | #Joint | #Revolute | #Prismatic |
|-----------|----------|-------|--------|-----------|------------|
| 168 | Faucet | 3 | 2 | 2 | 0 |
| 1280 | Faucet | 3 | 2 | 2 | 0 |
| 8961 | Door | 3 | 2 | 2 | 0 |
| 9016 | Door | 3 | 2 | 2 | 0 |
| 10489 | Refrigerator | 3 | 2 | 2 | 0 |
| 10655 | Refrigerator | 3 | 2 | 2 | 0 |
| 25493 | Table | 4 | 3 | 0 | 3 |
| 30666 | Table | 10 | 9 | 0 | 9 |
| 31249 | Table | 5 | 4 | 2 | 2 |
| 45194 | Storage Furniture | 5 | 4 | 2 | 2 |
| 45503 | Storage Furniture | 4 | 3 | 3 | 0 |
| 45612 | Storage Furniture | 7 | 6 | 4 | 2 |
| 47648 | Storage Furniture | 7 | 6 | 4 | 2 |
| 100481 | Bucket | 3 | 2 | 2 | 0 |
| 101284 | Eyeglasses | 3 | 2 | 2 | 0 |
| 101287 | Eyeglasses | 3 | 2 | 2 | 0 |
| 101808 | Oven | 3 | 2 | 2 | 0 |
| 101908 | Oven | 4 | 3 | 3 | 0 |
| 103015 | Window | 4 | 3 | 3 | 0 |
| 103811 | Printer | 7 | 6 | 0 | 6 |
| Average | — | 4.35 | 3.35 | 2.05 | 1.3 |

**Dual Quaternions for SE(3) Transformation**  A general rigid SE(3) transformation in 3D space consists of a rotation followed by a translation. A dual quaternion represents this combined operation within a single algebraic entity. Let the rotation be represented by a unit quaternion $\boldsymbol{q}_r$ and the translation by a vector $\boldsymbol{t}$. A point $\boldsymbol{p}$ in space, represented as a pure quaternion $\bar{\boldsymbol{p}} = (0, \boldsymbol{p})$, is transformed to a new point $\boldsymbol{p}'$ by first applying the rotation and then the translation: $\boldsymbol{p}' = \boldsymbol{q}_r \otimes \boldsymbol{p} \otimes \boldsymbol{q}_r^* + \boldsymbol{t}$, where $\boldsymbol{q}_r^*$ is the conjugate of $\boldsymbol{q}_r$ and $\otimes$ denotes the quaternion multiplication operation.

A dual quaternion $\boldsymbol{q}$ is defined as $\boldsymbol{q} = \boldsymbol{q}_r + \varepsilon \boldsymbol{q}_d$, where $\boldsymbol{q}_r$ is the real part, $\boldsymbol{q}_d$ is the dual part, and $\varepsilon$ is the dual unit with the property $\varepsilon^2 = 0$. Given the rotation quaternion $\boldsymbol{q}_r$ and translation $\boldsymbol{t}$, the dual part $\boldsymbol{q}_d$ could be calculated by: $\boldsymbol{q}_d = \frac{1}{2}(0, \boldsymbol{t}) \otimes \boldsymbol{q}_r$.

**Dual Quaternions for Articulated Transformation**  We apply the above principles to derive the specific formulas for prismatic and revolute joints at time $t$.

*Prismatic*: A prismatic joint executes a pure translation with no rotation, so that the real part is the unit quaternion $\boldsymbol{q}_r^t = (1, 0, 0, 0)$. Given the axis direction $\boldsymbol{d}$ and joint state $\theta^t$, its translation component is $\boldsymbol{t} = \theta^t \cdot \boldsymbol{d}$. Let $\bar{\boldsymbol{o}}^t = (0, \theta^t \cdot \boldsymbol{d})$, the dual part could be calculated by: $\boldsymbol{q}_d = \frac{1}{2}\bar{\boldsymbol{o}} \otimes \boldsymbol{q}_r^t$.

*Revolute*: A revolute joint executes a pure rotation, not about the world origin, but about the joint's origin point $\boldsymbol{o}$. Given the axis direction $\boldsymbol{d}$, axis origin $\boldsymbol{o}$ and joint state $\theta^t$ this "off-center" rotation is equivalent to a sequence of three operations: (1) translate the system so the pivot point $\boldsymbol{o}$ moves to the origin: $\bar{\boldsymbol{o}}^t = (0, \boldsymbol{o})$, $\boldsymbol{q}_{T_1} = 1 - \frac{\varepsilon}{2}\bar{\boldsymbol{o}}$); (2) perform the rotation around the origin: $\boldsymbol{q}_R = \boldsymbol{q}_r^t = (\cos\frac{\theta^t}{2}, \sin\frac{\theta^t}{2} \cdot \boldsymbol{d})$. (3) translate the system back: $\boldsymbol{q}_{T_2} = 1 + \frac{\varepsilon}{2}\bar{\boldsymbol{o}}$. The total transformation $\boldsymbol{q}^t$ is the product:

$$\boldsymbol{q}^t = \boldsymbol{q}_{T_2}\boldsymbol{q}_R\boldsymbol{q}_{T_1} = (1 + \frac{\varepsilon}{2}\bar{\boldsymbol{o}})\boldsymbol{q}_r^t(1 - \frac{\varepsilon}{2}\bar{\boldsymbol{o}}) = (1 + \frac{\varepsilon}{2}\bar{\boldsymbol{o}})(\boldsymbol{q}_r^t - \frac{\varepsilon}{2}\boldsymbol{q}_r^t\bar{\boldsymbol{o}}) = \boldsymbol{q}_r^t + \varepsilon\left(\frac{1}{2}\bar{\boldsymbol{o}}\boldsymbol{q}_r^t - \frac{1}{2}\boldsymbol{q}_r^t\bar{\boldsymbol{o}}\right),$$

where $\otimes$ is omitted for brevity. As a result, the real part and dual part are calculated as: $\boldsymbol{q}_r^t = (\cos\frac{\theta^t}{2}, \sin\frac{\theta^t}{2} \cdot \boldsymbol{d})$, $\boldsymbol{q}_d^t = \frac{1}{2}(\bar{\boldsymbol{o}} \otimes \boldsymbol{q}_r^t - \boldsymbol{q}_r^t \otimes \bar{\boldsymbol{o}})$.

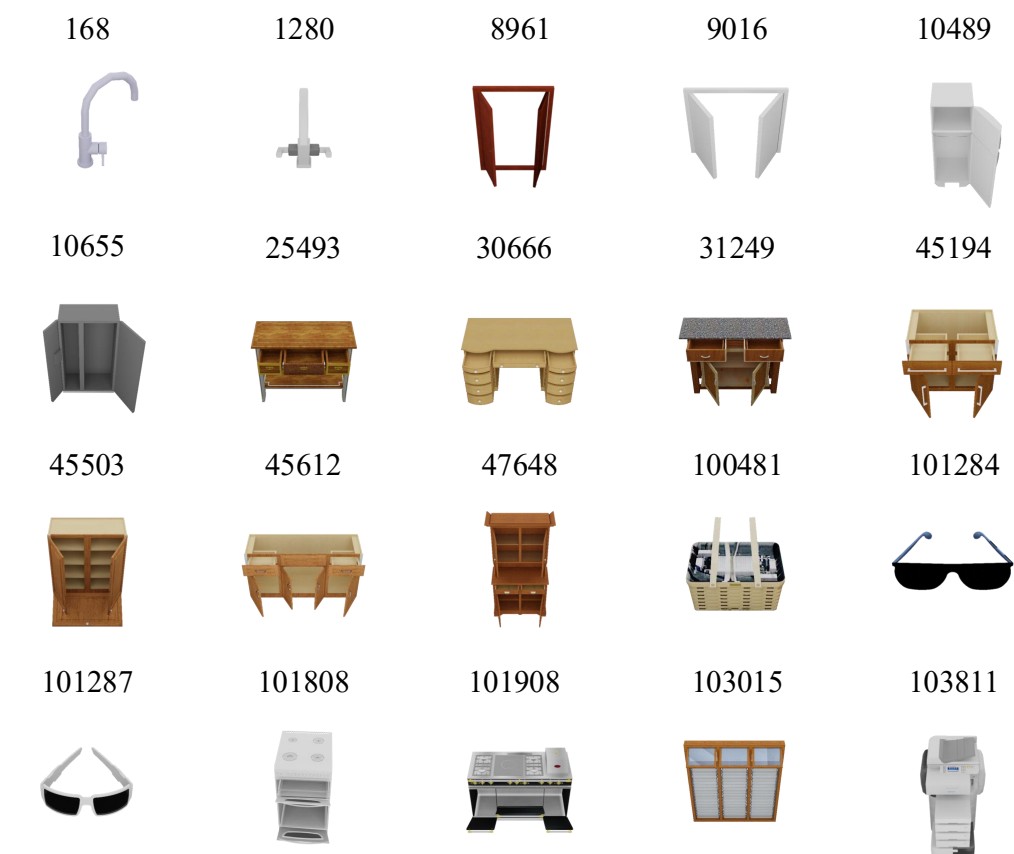

Figure A.1: Visualization of VideoArtGS-20 dataset.

## A.4 INITIALZATION AND OPTIMIZATION

**Inversed Deformation Field** Given a point position $x_i^{t_0}$ sampled from the trajectory $\{x_i^t\}_{t=1}^T$, we extend our part assignment module from canonical space to observation space to obtain the part assignment probabilities $m_i^{t_0}$ of $x_i^{t_0}$ at time $t_0$. Specifically, we deform the learnable centers $C_k = (p_k, V_k, \lambda_k)$ from canonical space to observation space by:

$$p_k^{t_0} = R_k^{t_0} \cdot p_k + t_k^{t_0}, \quad V_k^{t_0} = R_k^{t_0} \cdot V_k, \quad C_k^{t_0} = (p_k^{t_0}, V_k^{t_0}, \lambda_k), \tag{A.7}$$

where $R_k^{t_0}$ and $t_k^{t_0}$ is rotation matrix and translation vector derived from $q_k^{t_0}$. We replacing $C_k$ with $C_k^{t_0}$ in Eq. (2) and Eq. (3) to calculate $m_i^{t_0}$, then the canonical position $\hat{x}_i^c$ is calculated by:

$$q_i^{t_0} = \sum_{k=1}^K m_{ik}^{t_0} \cdot q_k^{t_0}, \quad \hat{x}_i^c = (R_i^{t_0})^{-1} \cdot (x_i^{t_0} - t_i^{t_0}), \tag{A.8}$$

where $R_i^{t_0}$ and $t_i^{t_0}$ is rotation matrix and translation vector derived from $q_i^{t_0}$.

**Training Configuration** We train deformation field $\mathcal{F}$ for 10K steps with loss $\mathcal{L}_{track} = \mathcal{L}_{o2o} + \mathcal{L}_{c2o}$ described in Eq. (4) and Eq. (5), taking 5-10 minutes per object. We train canonical Gaussians $\mathcal{G}^c$ for 20K steps with loss $\mathcal{L}_{render} = (1 - \lambda_{SSIM})\mathcal{L}_1 + \lambda_{SSIM}\mathcal{L}_{D\text{-}SSIM} + \mathcal{L}_D$, where $\lambda_{SSIM} = 0.2$ is used in experiments. This stage takes about 4 minutes per object. We jointly optimize the canonical Gaussians and deformation field for 20K steps with $\mathcal{L} = \mathcal{L}_I + \mathcal{L}_D + \lambda_{c2o}\mathcal{L}_{c2o}$, where $\lambda_{c2o} = 0.5$. This state takes 10-20 minutes per object.

## A.5 JOINT TYPE PREDICTION USING GPT-4O

Inspired by SINGAPO (Liu et al., 2025), we use GPT-4o to predict the number of joints and joint types. We input the video and a step-by-step instruction to make GPT-4o understand the articulated objects. The version of GPT-4o used in our experiments is gpt-4o-2024-11-20. The instruction is:

```
💻 System:

The user will provide a sequence of pictures of articulated objects, some of which shows how human interact with it. The system will analyze:

1. Identify the articulated object in each picture.
2. Identify the static part of this articulated object.
3. Identify all movable parts of the object and output the composed information in the form of one json file.

Example answer:
The articulated object is a carbinet.
The static part is the cabinet base.
The movable parts are the top-drawer, bottom-drawer, and middle-drawer.
The joint type is a slider.
The output json file is:
[
    {
        "id": 0,
        "name": "cabinet_base",
        "joint": "heavy",
        "parent": -1
    },
    {
        "id": 1,
        "name": "top_drawer",
        "joint": "slider",
        "parent": 0
    },·
    {
        "id": 2,
        "name": "bottom_drawer",
        "joint": "slider",
        "parent": 1
    },
    {
        "id": 3,
        "name": "middle_drawer",
        "joint": "slider",
        "parent": 1
    }
]

👤 User:

Analyze the picture of an articulated object and output the information in the form of a json file.
```

Figure A.2: Prompt for GPT-4o to predict the number of joints and joint types.

## B LIMITATIONS

Our method, while effective, has limitations that open avenues for future research.

**Dependency on Upstream Perception Models.** The final quality of our reconstruction is inherently dependent on the accuracy of the upstream models used for perception. Our pipeline first relies on a monocular depth and camera pose estimator (e.g., VGGT). Subsequently, a pre-trained tracking model (e.g., TAPIP3D) generates 3D motion tracks. If the depth or camera pose estimates contain significant errors, the resulting 3D tracks will be noisy and fail to capture the object's true rigid-body motion. This can lead to failures in our downstream fitting and clustering steps, resulting in distorted geometry or incorrect joint estimation. However, as this is a rapidly advancing field, we anticipate that progress in visual foundation models and tracking models will continue to mitigate this dependency.

**Canonical Gaussian Initialization.** Our current framework assumes that the input video begins with a short sequence (N frames) where the scene is static. This segment is crucial for initializing the canonical Gaussian representation of the object's geometry. While this assumption is practical for data captured by a user (self-shot), it restricts the method's applicability to in-the-wild videos from the internet, which often begin with immediate motion. Relaxing this condition is non-trivial, as it makes the ill-posed problem of disentangling geometry from motion even more challenging. A promising direction for future work is to incorporate powerful generative priors. Such models could help infer a plausible canonical shape even from a video with continuous motion, thereby enabling reconstruction from arbitrary monocular inputs.

Table A.2: **Detailed results on Video2Articulation-S dataset**. Following Video2Articulation(Peng et al., 2025), if the method failed, we assign a 90° angle error for the joint axis, and a 100cm error for the other metrics. "Diff." denotes the results of Video2Articulation minus ours, which demonstrate the improvements of VideoArtGS. The **best results** are highlighted in bold.

| Metric | CD-s (cm) | | | CD-w (cm) | | | CD-m (cm) | | | Axis (○) | | | Position (cm) | | |
|---|---|---|---|---|---|---|---|---|---|---|---|---|---|---|---|
| Method | V2A | Ours | Diff. | V2A | Ours | Diff. | V2A | Ours | Diff. | V2A | Ours | Diff. | V2A | Ours | Diff. |
| Box | 1.56 | **0.03** | -1.53 | 1.04 | **0.07** | -0.96 | 0.34 | **0.10** | -0.24 | 0.06 | **0.06** | 0.00 | 2.31 | **0.17** | -2.14 |
| Dishwasher | 0.74 | **0.37** | -0.37 | 0.58 | **0.30** | -0.28 | 3.34 | **0.14** | -3.20 | 3.27 | **0.11** | -3.16 | 11.93 | **0.39** | -11.54 |
| Laptop | 7.30 | **0.15** | -7.15 | 0.09 | **0.09** | 0.00 | 0.13 | **0.10** | -0.03 | 8.40 | **0.06** | -8.35 | 3.48 | **0.22** | -3.26 |
| Microwave | 2.84 | **0.35** | -2.49 | 2.06 | **0.31** | -1.74 | 0.20 | **0.07** | -0.13 | 0.40 | **0.26** | -0.14 | 9.49 | **0.10** | -9.39 |
| Refrigerator | 22.91 | **0.28** | -22.63 | 22.66 | **0.16** | -22.50 | 22.05 | **0.49** | -21.56 | 18.91 | **0.67** | -18.24 | 23.08 | **0.30** | -22.78 |
| Scissors | 10.05 | **0.16** | -9.88 | 0.02 | **0.02** | 0.00 | 8.52 | **0.04** | -8.47 | 30.21 | **0.32** | -29.90 | 5.86 | **0.23** | -5.63 |
| Stapler | 64.07 | **0.37** | -63.70 | 50.13 | **0.27** | -49.85 | 51.26 | **0.19** | -51.07 | 46.07 | **0.05** | -46.01 | 50.19 | **1.83** | -48.36 |
| StorageFurniture | **0.61** | 0.68 | 0.07 | 0.48 | **0.48** | 0.00 | 10.39 | **3.02** | -7.37 | 4.47 | **0.14** | -4.33 | 0.61 | **0.01** | -0.60 |
| Table | 0.40 | **0.35** | -0.04 | 0.43 | **0.32** | -0.11 | 19.21 | **1.56** | -17.65 | 17.22 | **0.44** | -16.78 | 3.01 | **0.07** | -2.94 |
| USB | 1.75 | **0.43** | -1.32 | 0.24 | **0.19** | -0.05 | 4.27 | **0.95** | -3.33 | 11.29 | **0.18** | -11.12 | 3.95 | **0.05** | -3.90 |
| WashingMachine | 2.20 | **0.82** | -1.37 | 1.93 | **0.76** | -1.17 | 4.32 | **0.09** | -4.23 | 13.03 | **1.29** | -11.74 | 5.28 | **1.89** | -3.39 |

Table A.3: **Detailed results on VideoArtGS-20 dataset**. Following Video2Articulation(Peng et al., 2025), if the method failed, we assign a 90° angle error for the joint axis, and a 100cm error for the other metrics. "Diff." denotes the results of Video2Articulation minus ours, which demonstrate the improvements of VideoArtGS. The **best results** are highlighted in bold.

| Metric | CD-s (cm) | | | CD-w (cm) | | | CD-m (cm) | | | Axis (○) | | | Position (cm) | | |
|---|---|---|---|---|---|---|---|---|---|---|---|---|---|---|---|
| Method | V2A | Ours | Diff. | V2A | Ours | Diff. | V2A | Ours | Diff. | V2A | Ours | Diff. | V2A | Ours | Diff. |
| 100481 | 2.63 | **0.11** | -2.52 | **0.05** | 0.11 | 0.07 | 1.43 | **0.03** | -1.40 | 43.93 | **0.06** | -43.88 | 35.92 | **0.00** | -35.91 |
| 101284 | 7.28 | **0.01** | -7.28 | 0.02 | **0.01** | -0.01 | 76.58 | **0.00** | -76.58 | 14.91 | **0.18** | -14.73 | 29.31 | **0.13** | -29.18 |
| 101287 | 10.16 | **0.01** | -10.15 | 0.01 | **0.01** | 0.00 | 24.70 | **0.00** | -24.70 | 9.07 | **0.20** | -8.87 | 28.55 | **0.03** | -28.52 |
| 101808 | 0.89 | **0.10** | -0.79 | **0.09** | 0.10 | 0.01 | 13.78 | **0.01** | -13.76 | 44.95 | **3.79** | -41.16 | 35.94 | **0.02** | -35.92 |
| 101908 | 2.02 | **0.10** | -1.93 | **0.05** | 0.09 | 0.04 | 19.53 | **0.01** | -19.52 | 78.44 | **0.14** | -78.29 | 65.14 | **0.11** | -65.03 |
| 103015 | 100.00 | **0.27** | -99.73 | 100.00 | **0.24** | -99.76 | 100.00 | **0.01** | -99.99 | 54.74 | **0.07** | -54.66 | 49.06 | **0.12** | -48.93 |
| 103811 | 4.51 | **0.53** | -3.98 | **0.30** | 0.43 | 0.13 | 18.74 | **2.65** | -16.09 | 80.81 | **0.22** | -80.59 | 0.00 | **0.00** | 0.00 |
| 10489 | 3.60 | **0.05** | -3.55 | 0.07 | **0.06** | -0.01 | 60.79 | **0.01** | -60.78 | 68.98 | **0.08** | -68.89 | 133.32 | **0.18** | -133.14 |
| 10655 | 3.05 | **0.06** | -2.99 | 0.10 | **0.08** | -0.02 | 0.08 | **0.01** | -0.07 | 7.71 | **0.03** | -7.68 | 35.16 | **0.20** | -34.96 |
| 1280 | 0.56 | **0.11** | -0.45 | **0.02** | 0.03 | 0.01 | 6.07 | **0.03** | -6.04 | 74.84 | **0.68** | -74.15 | 48.53 | **0.32** | -48.21 |
| 168 | 39.67 | **2.68** | -36.99 | **0.02** | 0.04 | 0.02 | 17.23 | **0.43** | -16.80 | 53.59 | **0.32** | -53.27 | 10.59 | **0.22** | -10.36 |
| 25493 | 0.75 | **0.11** | -0.63 | 0.09 | **0.06** | -0.03 | 65.98 | **0.18** | -65.79 | 53.67 | **0.14** | -53.53 | 0.00 | **0.00** | 0.00 |
| 30666 | 1.19 | **0.22** | -0.97 | **0.12** | 0.17 | 0.05 | 37.51 | **1.17** | -36.34 | 66.94 | **0.14** | -66.80 | 0.00 | **0.00** | 0.00 |
| 31249 | 1.51 | **0.11** | -1.41 | 0.09 | **0.09** | 0.00 | 11.98 | **0.22** | -11.76 | 40.68 | **0.08** | -40.59 | 3.69 | **0.02** | -3.68 |
| 45194 | 4.60 | **0.09** | -4.51 | **0.09** | 0.10 | 0.01 | 41.50 | **0.01** | -41.49 | 47.82 | **0.11** | -47.71 | 43.39 | **0.03** | -43.36 |
| 45503 | 3.11 | **0.07** | -3.04 | **0.07** | 0.08 | 0.01 | 21.32 | **0.01** | -21.31 | 62.82 | **0.03** | -62.79 | 34.59 | **0.11** | -34.48 |
| 45612 | 1.87 | **0.06** | -1.81 | **0.06** | 0.07 | 0.02 | 28.81 | **0.02** | -28.79 | 48.19 | **0.13** | -48.06 | 16.74 | **0.05** | -16.70 |
| 47648 | 0.69 | **0.06** | -0.64 | 0.05 | **0.06** | 0.00 | 24.90 | **0.33** | -24.57 | 48.90 | **0.18** | -48.72 | 37.10 | **0.06** | -37.04 |
| 8961 | 4.92 | **0.02** | -4.90 | 0.03 | **0.03** | 0.00 | 27.25 | **0.02** | -27.23 | 1.05 | **0.02** | -1.03 | 84.21 | **0.05** | -84.16 |
| 9016 | 11.38 | **0.02** | -11.36 | 0.03 | **0.03** | 0.00 | 14.42 | **0.02** | -14.40 | 75.53 | **0.10** | -75.43 | 49.52 | **0.35** | -49.17 |
| Average | 10.22 | **0.24** | -9.98 | 5.07 | **0.09** | -4.97 | 30.63 | **0.26** | -30.37 | 48.88 | **0.34** | -48.54 | 37.04 | **0.10** | -36.93 |

**Reliance on Motion for Part Segmentation.** Our approach infers part segmentation exclusively from motion cues by clustering the derived 3D tracks. This reliance on dynamics places high demands on the tracking quality and can be fragile for objects with many parts or for parts that exhibit very subtle relative motion. In such challenging cases, the segmentation quality can degrade, leading to incorrectly merged or split components. A valuable future direction is to augment our motion-based clustering with appearance-based priors from pre-trained foundation models. For instance, integrating semantic features from DINOv2 (Oquab et al., 2023) or segmentation masks from models like SAM (Kirillov et al., 2023) could provide a powerful, independent signal for identifying object parts, making the segmentation process significantly more robust.

# C ADDITIONAL EXPERIMENT RESULTS AND ANALYSIS

## C.1 DETAILED RESULTS ON VIDEO2ARTICULATION-S AND VIDEOARTGS-20

We provide detailed results on the Video2Articulation-S and VideoArtGS-20 dataset. As shown in Appendix C.1 and Tab. A.3, we observe consistent improvements across all categories in both Video2Articulation-S and VideoArtGS-20 datasets.

Table A.4: **Tracking ability comparison on VideoArtGS-20 dataset**. "Diff." denotes the results of TAPIP3D minus ours, which demonstrate the improvements of VideoArtGS. The **best results** are highlighted in bold.

| Metric | EPE (m) ↓ | | | $\delta_{0.05}$ ↑ | | | $\delta_{0.10}$ ↑ | | |
|---|---|---|---|---|---|---|---|---|---|
| Method | TAPIP3D | Ours | Diff. | TAPIP3D | Ours | Diff. | TAPIP3D | Ours | Diff. |
| 100481 | 0.16 | **0.12** | -0.04 | 59.04 | **66.95** | 7.92 | 65.40 | **72.76** | 7.37 |
| 101284 | 0.03 | **0.01** | -0.02 | 88.99 | **95.96** | 6.98 | 90.46 | **96.75** | 6.29 |
| 101287 | 0.08 | **0.02** | -0.06 | 70.11 | **91.73** | 21.62 | 74.16 | **93.16** | 19.00 |
| 101808 | 0.13 | **0.06** | -0.07 | 57.83 | **76.65** | 18.82 | 64.66 | **81.31** | 16.65 |
| 101908 | 0.14 | **0.06** | -0.08 | 58.73 | **81.16** | 22.43 | 64.64 | **84.38** | 19.74 |
| 103015 | 0.02 | **0.01** | -0.02 | 89.70 | **93.38** | 3.68 | 93.35 | **96.22** | 2.87 |
| 103811 | 0.16 | **0.09** | -0.08 | 65.16 | **80.87** | 15.70 | 67.86 | **81.15** | 13.29 |
| 10489 | 0.32 | **0.14** | -0.18 | 45.49 | **75.72** | 30.23 | 50.88 | **78.84** | 27.95 |
| 10655 | 0.30 | **0.14** | -0.15 | 35.31 | **66.00** | 30.69 | 43.81 | **69.36** | 25.55 |
| 1280 | 0.04 | **0.02** | -0.02 | 83.63 | **91.12** | 7.49 | 88.27 | **94.07** | 5.80 |
| 168 | 0.43 | **0.32** | -0.11 | 43.91 | **52.82** | 8.91 | 51.29 | **58.39** | 7.10 |
| 25493 | 0.08 | **0.04** | -0.04 | 72.33 | **78.55** | 6.21 | 76.59 | **83.49** | 6.90 |
| 30666 | 0.12 | **0.06** | -0.05 | 59.75 | **80.04** | 20.29 | 64.98 | **80.28** | 15.30 |
| 31249 | 0.12 | **0.07** | -0.05 | 60.21 | **76.67** | 16.47 | 63.36 | **78.54** | 15.17 |
| 45194 | 0.23 | **0.12** | -0.10 | 42.23 | **70.65** | 28.42 | 48.28 | **71.34** | 23.06 |
| 45503 | 0.19 | **0.10** | -0.09 | 42.32 | **69.75** | 27.43 | 48.46 | **72.88** | 24.41 |
| 45612 | 0.16 | **0.08** | -0.09 | 35.50 | **66.47** | 30.98 | 43.17 | **69.60** | 26.43 |
| 47648 | 0.09 | **0.03** | -0.06 | 58.05 | **81.46** | 23.42 | 66.89 | **82.98** | 16.10 |
| 8961 | 0.18 | **0.15** | -0.02 | 22.74 | **29.21** | 6.47 | 37.46 | **46.51** | 9.05 |
| 9016 | **0.16** | 0.26 | 0.10 | 20.36 | **35.27** | 14.91 | 37.47 | **47.54** | 10.06 |
| Average | 0.16 | **0.09** | -0.06 | 55.57 | **73.02** | 17.45 | 62.07 | **76.98** | 14.91 |

## C.2    TRACKING IMPROVEMENT

Tracks from TAPIP3D (Zhang et al., 2025) maintain noisy and inaccurate trajectories. Our pipeline filters noise and refines the tracks, enabling more accurate learning of dynamic and articulation parameters. We add track-quality metrics to demonstrate our method's track-correction capability.

**Evaluation protocol.** Following Shape of Motion (Wang et al., 2024a), we use the 3D end-point-error (EPE), which measures the Euclidean distance between ground truth and predicted 3D tracks. We also report the percentage of points falling within given thresholds of the ground truth: $\delta_{0.05} = 5cm$ and $\delta_{0.10} = 10cm$. Given noisy tracks from TAPIR3D (Zhang et al., 2025), we use our motion prior guidance pipeline to filter noise and then input the filtered tracks as query points to our optimized model, calculating new tracks using our learned deformation field. Ground truth tracks are obtained by deforming the query points with ground truth joint parameters and states at each time step. Each query point inherits the part label of its nearest vertex on the ground truth meshes.

**Results.** As shown in Tab. A.4, our method consistently improves upon TAPIP3D (Zhang et al., 2025) across all metrics. These improvements are particularly significant for challenging objects (e.g. 47648, 30666) with complex articulated motion.

## C.3    SENSITIVITY ANALYSIS OF FITTING THRESHOLD $\epsilon_l/\epsilon_c$

Our method uses equal thresholds for line/circle fitting ($\epsilon_l/\epsilon_c$). Throughout our main experiments, we adopt $\epsilon_l = \epsilon_c = 0.01$. We provide comprehensive experimental results across different threshold values in Tab. A.5 and Tab. A.6.

Importantly, our method does not require filtering out all noise. Our hybrid center-grid assignment module is learnable—its parameters are optimized via gradient descent during training, which naturally corrects initialization errors from tracking noise. The initialization only needs to provide reasonable starting parameters for robust optimization.

As shown in Tab. A.5 and Tab. A.6, the initialization joint parameter errors exhibit minimal variation across different thresholds, and the differences in optimized metrics are similarly small. A clear trend emerges: smaller thresholds filter out more noise, yielding more accurate initial joint parameters. However, excessively small thresholds (e.g., 0.005) cause most trajectories to be classified as noise, leading to significant performance degradation on certain objects (e.g., object 168). Our experiments demonstrate robust performance across reasonable threshold variations.

Table A.5: Initialization joint parameter error with different fitting threshold $\epsilon_l/\epsilon_c$ on VideoArtGS-20 dataset. The **best results** are highlighted in bold.

| Metric | Axis (∘) | | | | | Position (cm) | | | | |
|---|---|---|---|---|---|---|---|---|---|---|
| $\epsilon_l/\epsilon_c$ | 0.100 | 0.040 | 0.020 | 0.010 | 0.005 | 0.100 | 0.040 | 0.020 | 0.010 | 0.005 |
| 100481 | 5.45 | 5.45 | 5.70 | 5.80 | **4.45** | 0.16 | 0.16 | **0.15** | 1.52 | 6.92 |
| 101284 | **0.86** | 0.86 | 0.93 | 0.93 | 2.14 | **1.34** | **1.34** | 1.42 | 1.70 | 2.95 |
| 101287 | **1.07** | 1.07 | 1.14 | 1.36 | 3.07 | 0.35 | 0.35 | 0.22 | **0.21** | 1.21 |
| 101808 | 7.76 | 7.76 | 7.78 | 7.70 | **7.04** | 1.00 | 1.00 | 1.00 | 1.02 | **0.92** |
| 101908 | 1.21 | 1.21 | 1.15 | 1.42 | **0.89** | 1.20 | 1.20 | 1.60 | 1.14 | **0.32** |
| 103015 | **5.89** | **5.89** | 5.90 | 6.14 | 6.13 | **1.25** | **1.25** | **1.25** | 1.26 | 1.36 |
| 103811 | 0.69 | 0.54 | 0.65 | 0.57 | **0.38** | 0.00 | 0.00 | 0.00 | 0.00 | 0.00 |
| 10489 | 0.99 | 0.99 | 0.94 | 1.02 | **0.50** | 0.22 | **0.22** | 0.32 | 0.40 | 0.39 |
| 10655 | **1.48** | 1.50 | 1.56 | 1.68 | 2.18 | 0.54 | **0.51** | 0.52 | 0.54 | 25.51 |
| 1280 | 4.88 | 4.88 | 4.88 | 5.58 | **2.48** | 1.23 | 1.23 | 1.23 | 1.58 | **0.72** |
| 168 | **1.54** | **1.54** | 1.55 | 1.65 | 44.25 | 1.41 | 1.41 | 1.40 | **1.32** | 3.38 |
| 25493 | 0.34 | 0.34 | 0.34 | 0.37 | **0.31** | 0.00 | 0.00 | 0.00 | 0.00 | 0.00 |
| 30666 | 1.10 | 1.37 | 0.96 | 0.91 | **0.80** | 0.00 | 0.00 | 0.00 | 0.00 | 0.00 |
| 31249 | 2.50 | 2.49 | 2.16 | 1.08 | **0.50** | 0.12 | **0.12** | **0.12** | 0.16 | 0.25 |
| 45194 | 1.75 | 1.66 | 1.37 | 0.66 | **0.25** | 0.22 | 0.22 | **0.21** | 0.26 | 0.34 |
| 45503 | 1.24 | 1.23 | 1.23 | **1.22** | 2.95 | 0.10 | 0.10 | 0.10 | 0.11 | **0.08** |
| 45612 | 3.14 | 3.15 | 2.77 | 1.91 | **1.72** | 0.21 | 0.21 | 0.20 | 0.25 | **0.16** |
| 47648 | 6.16 | 4.95 | 2.83 | 2.89 | **2.09** | 0.19 | 0.19 | 0.19 | **0.17** | 0.24 |
| 8961 | 0.96 | 0.96 | 0.97 | 0.98 | **0.74** | 0.42 | 0.42 | 0.42 | 0.41 | **0.33** |
| 9016 | 1.38 | 1.38 | 1.51 | 1.44 | **0.80** | 0.18 | **0.16** | 0.98 | 0.60 | 2.84 |
| Average | 2.52 | 2.46 | 2.32 | **2.26** | 4.18 | 0.51 | **0.50** | 0.57 | 0.63 | 2.39 |

Table A.6: Optimized results with different fitting threshold $\epsilon_l/\epsilon_c$ on VideoArtGS-20 dataset. We ignore the CD-w metric because it changes almost imperceptibly with different $\epsilon_l/\epsilon_c$. The **best results** are highlighted in bold.

| Metric | CD_s (cm) | | | | CD_m (cm) | | | | Axis (∘) | | | | Position (cm) | | | |
|---|---|---|---|---|---|---|---|---|---|---|---|---|---|---|---|---|
| $\epsilon_l/\epsilon_c$ | V2A | 0.100 | 0.010 | 0.005 | V2A | 0.100 | 0.010 | 0.005 | V2A | 0.100 | 0.010 | 0.005 | V2A | 0.100 | 0.010 | 0.005 |
| 100481 | 2.63 | **0.11** | **0.11** | 0.37 | 1.43 | **0.02** | 0.03 | 14.07 | 43.93 | **0.01** | 0.06 | 12.37 | 35.92 | 0.02 | **0.00** | 2.53 |
| 101284 | 7.28 | **0.01** | **0.01** | **0.01** | 76.58 | **0.00** | **0.00** | **0.00** | 14.91 | **0.14** | 0.18 | 0.35 | 29.31 | **0.10** | 0.13 | 0.15 |
| 101287 | 10.16 | **0.01** | **0.01** | **0.01** | 24.70 | **0.00** | **0.00** | **0.00** | 9.07 | 0.25 | **0.20** | 0.28 | 28.55 | 0.06 | **0.03** | 0.07 |
| 101808 | 0.89 | **0.10** | **0.10** | **0.10** | 13.78 | **0.01** | **0.01** | **0.01** | 44.95 | **3.79** | **3.79** | 3.80 | 35.94 | 0.09 | **0.02** | 0.09 |
| 101908 | 2.02 | **0.10** | **0.10** | **0.10** | 19.53 | **0.01** | **0.01** | **0.01** | 78.44 | 0.12 | 0.14 | **0.10** | 65.14 | **0.08** | 0.11 | 0.10 |
| 103015 | 100.00 | 0.20 | 0.27 | **0.18** | 100.00 | **0.01** | **0.01** | **0.01** | 54.74 | 0.13 | **0.07** | 0.13 | 49.06 | 0.24 | **0.12** | 0.23 |
| 103811 | 4.51 | 0.49 | 0.53 | **0.43** | 18.74 | 4.33 | **2.65** | 21.53 | 80.81 | **0.17** | 0.22 | 1.24 | 0.00 | **0.00** | **0.00** | **0.00** |
| 10489 | 3.60 | **0.05** | **0.05** | **0.05** | 60.79 | **0.01** | **0.01** | **0.01** | 68.98 | **0.08** | **0.08** | 0.09 | 133.32 | 0.29 | **0.18** | 0.27 |
| 10655 | 3.05 | **0.06** | **0.06** | 2.40 | 0.08 | **0.01** | **0.01** | 132.42 | 7.71 | **0.01** | 0.03 | 0.04 | 35.16 | **0.16** | 0.20 | 17.51 |
| 1280 | 0.56 | 0.09 | 0.11 | **0.08** | 6.07 | 0.08 | **0.03** | 0.12 | 74.84 | 0.67 | 0.68 | 1.05 | 48.53 | 0.31 | 0.32 | **0.30** |
| 168 | 39.67 | 3.13 | **2.68** | 3.74 | 17.23 | **0.25** | 0.43 | 92.35 | 53.59 | 0.35 | **0.32** | 45.11 | 10.59 | 0.25 | **0.22** | 0.52 |
| 25493 | 0.75 | **0.11** | **0.11** | 0.16 | 65.98 | **0.15** | 0.18 | 0.16 | 53.67 | **0.11** | 0.14 | **0.11** | 0.00 | **0.00** | **0.00** | **0.00** |
| 30666 | 1.19 | **0.22** | **0.22** | **0.22** | 37.51 | 1.20 | **1.17** | 13.36 | 66.94 | **0.13** | 0.14 | 0.66 | 0.00 | **0.00** | **0.00** | **0.00** |
| 31249 | 1.51 | **0.10** | 0.11 | 0.11 | 11.98 | **0.17** | 0.22 | 0.24 | 40.68 | **0.07** | 0.08 | **0.07** | 3.69 | **0.00** | 0.02 | 0.01 |
| 45194 | 4.60 | 0.15 | **0.09** | 0.12 | 41.50 | 0.14 | **0.01** | 0.09 | 47.82 | 0.25 | **0.11** | 0.23 | 43.39 | 15.82 | **0.03** | 15.59 |
| 45503 | 3.11 | **0.07** | **0.07** | **0.07** | 21.32 | **0.01** | **0.01** | **0.01** | 62.82 | **0.02** | 0.03 | **0.02** | 34.59 | **0.10** | 0.11 | **0.10** |
| 45612 | 1.87 | **0.06** | **0.06** | **0.06** | 28.81 | **0.01** | 0.02 | **0.01** | 48.19 | 0.12 | 0.13 | **0.11** | 16.74 | **0.04** | 0.05 | 0.05 |
| 47648 | 0.69 | 0.07 | **0.06** | 0.07 | 24.90 | 16.92 | **0.33** | 16.94 | 48.90 | 39.79 | **0.18** | 36.81 | 37.10 | 3.21 | **0.06** | 3.17 |
| 8961 | 4.92 | **0.02** | **0.02** | **0.02** | 27.25 | **0.02** | **0.02** | **0.02** | 1.05 | **0.02** | **0.02** | **0.02** | 84.21 | **0.03** | 0.05 | 0.05 |
| 9016 | 11.38 | **0.02** | **0.02** | **0.02** | 14.42 | 0.02 | 0.02 | **0.01** | 75.53 | 0.10 | 0.10 | **0.09** | 49.52 | 0.34 | 0.35 | **0.32** |
| Average | 10.22 | 0.26 | **0.24** | 0.41 | 30.63 | 1.17 | **0.26** | 14.57 | 48.88 | 2.32 | **0.34** | 5.13 | 37.04 | 1.06 | **0.10** | 2.05 |

## C.4 EFFICIENCY COMPARISON

We provide runtime and GPU memory comparison with a single NVIDIA RTX 3090 GPU in Tab. A.7. Our method achieves a favorable balance between efficiency and reconstruction quality. While ArticulateAnything is faster, it uses GPT to predict joint parameters rather than reconstructing from video, making direct comparison less meaningful. Compared to Video2Articulation, our approach reduces GPU memory requirements and significantly improves efficiency. Video2Articulation requires 20-30 minutes per joint, meaning multi-joint objects can take hours to process, whereas our method completes reconstruction for objects with multiple parts in 15-40 minutes total. This efficiency gain stems from our joint optimization framework, which simultaneously reconstructs all articulated parts rather than processing each joint sequentially.

Table A.7: Efficiency Comparison.

| Method | GPU Memory | Runtime |
|---|---|---|
| ArticulateAnything | N/A | 2-5 minutes |
| Video2Articulation | 24GB | 20-30 minutes per joint |
| Ours | 12GB | 15-40 minutes |

Table A.8: Failure cases of GPT-4o. '1r1p' means 1 revolute joint and 1 prismatic joint.

| Dataset | Object ID | Category | GT Joint | Pred. Joint |
|---|---|---|---|---|
| Video2Articulation-S | 19898 | Table | 1r | 1p |
| Video2Articulation-S | 22433 | Table | 1r | 1p |
| VideoArtGS-20 | 30666 | Table | 9p | 7p |
| VideoArtGS-20 | 103811 | Printer | 6p | 1r1p |
| VideoArtGS-20 | 1280 | Faucet | 2r | 3r |

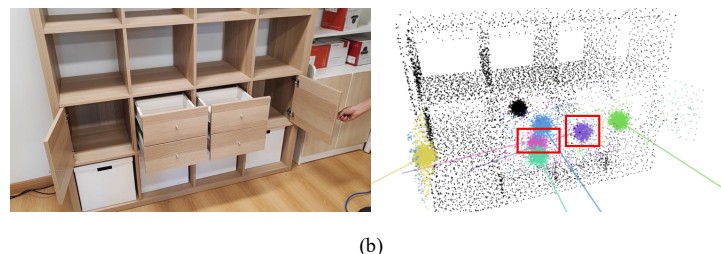

(a) (b)

Figure A.3: **Failure cases**. We illustrate failure cases of our VideoArtGS.

## C.5 FAILURE CASE ANALYSIS

**GPT-4o Prediction**    We provide failure cases of GPT-4o prediction in Tab. A.8. GPT-4o made 2 incorrect predictions on the 73 videos in Video2Articulation-S (wrong joint types) and 3 incorrect predictions on VideoArtGS-20 (wrong number of parts or joint types). The model tends to misclassify joint types when the range of motion is limited and underestimates the number of parts in objects with multiple joints. Notably, we tried alternative methods for automatically detecting the number of parts in the early stages of our experiments, including clustering and detection-based approaches. However, these methods exhibited weaker generalization than GPT-4o. Their reliance on additional assumptions makes them more prone to failure on novel data.

**Imperfect Part Segmentation**    As shown in Fig. A.3 (a), when multiple components are spatially close and share identical motion patterns (e.g., prismatic joints moving in the same direction), our method may fail to correctly segment them. As discussed in Appendix B, this limitation arises from the lack of semantic information in our current approach. Integrating semantic segmentation models such as SAM (Kirillov et al., 2023) could potentially address this issue.

**Clustering Error**    As shown in Fig. A.3 (b), when objects have numerous parts and tracks contain substantial noise, our method may produce incorrect initial centers. Large initialization errors are difficult to correct during optimization, as the model tends to converge to local minima. Addressing this limitation requires more robust methods for discovering part centers, such as incorporating semantic information to help with clustering.

## D    LLM USAGE STATEMENT

The authors acknowledge the use of Large Language Models (LLM) in the preparation of this paper. LLM was used to assist with improving writing clarity and grammar checking throughout the document. All AI-generated suggestions were carefully reviewed, modified as necessary, and validated by the authors. The core research contributions, experimental design, data analysis, and scientific conclusions are entirely the original work of the authors.

## E    ADDITIONAL QUALITATIVE RESULTS

We provide additional qualitative results in the following pages.

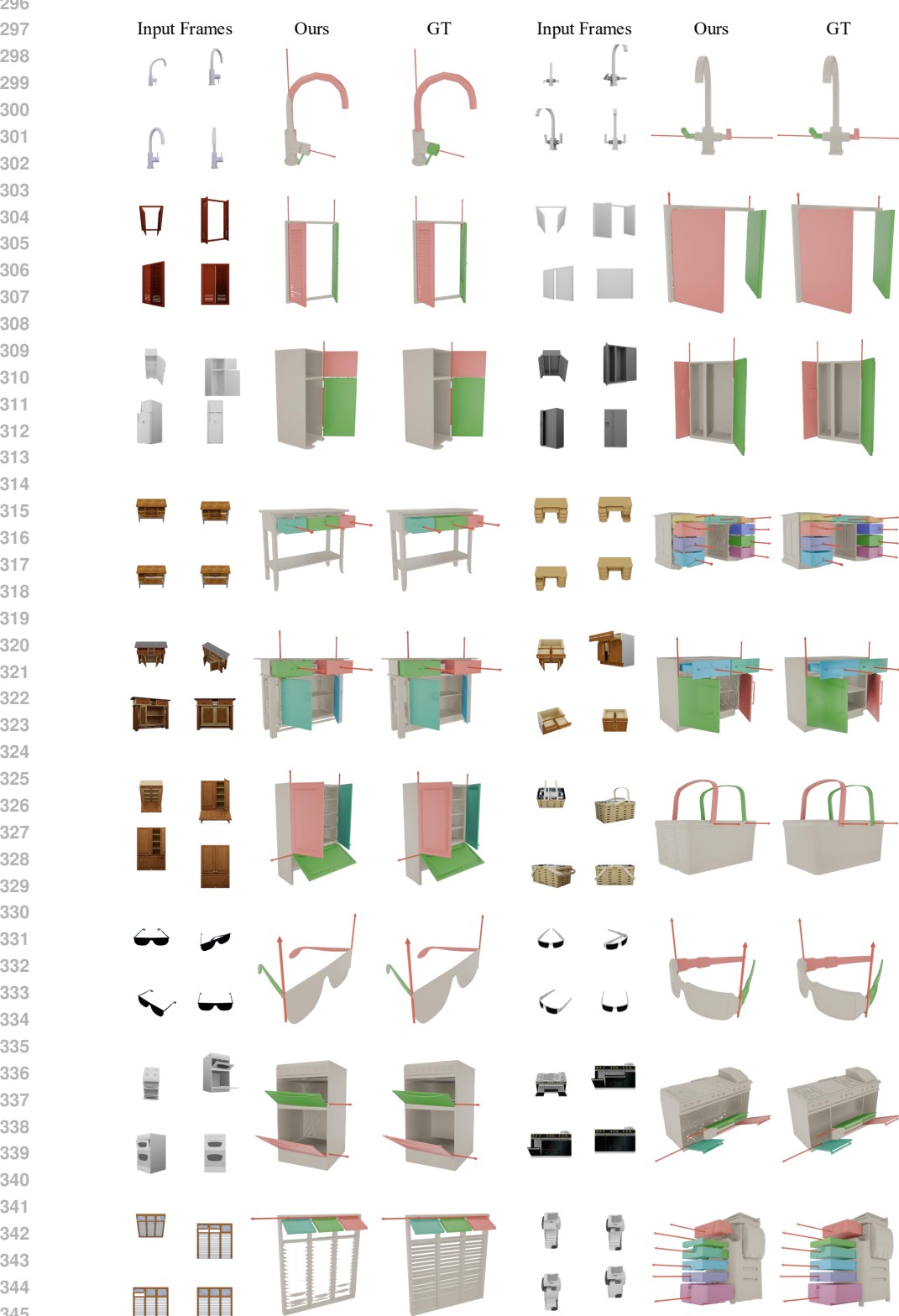

Figure A.4: **Additional qualitative results on VideoArtGS-20.**

