# OpenReview forum: "VideoArtGS: Building Digital Twins of Articulated Objects from Monocular Video"
_ICLR.cc/2026/Conference — Submitted to ICLR 2026_

### Official Review · Reviewer_PkAR · 2025-10-26

**Soundness:** 2
**Presentation:** 2
**Contribution:** 2
**Rating:** 4
**Confidence:** 5

**Summary:**

This paper tackles the problem of reconstructing articulated objects from monocular video, which requires simultaneously estimating geometry, part segmentation, and articulation parameters. The challenge lies in disentangling object geometry from part dynamics when both camera and object parts move together. The paper introduces VideoArtGS, which reconstructs digital twins through: (1) a motion prior guidance pipeline that analyzes 3D tracks and initializes articulation parameters, and (2) a hybrid center-grid part assignment module for accurate part motion capture.

**Strengths:**

1. The paper effectively addresses articulated object modeling from monocular video by incorporating motion priors from pre-trained tracking models to resolve reconstruction ambiguities.

2. The hybrid center-grid part assignment module combined with the motion prior guidance pipeline offers an end-to-end framework that handles both initialization and accurate part motion modeling.

**Weaknesses:**

1. The VideoArtGS-20 testing set should not be listed as a contribution, as it simply consists of 20 rendered objects from the existing PartNet-Mobility dataset without significant added value.

2. The reconstructions lack smoothness and exhibit really coarse surfaces, particularly evident in real-world examples in Figure 4 (e.g., the laptop and chair). The geometry appears very noisy.

3. Using GPT-4o to predict joint number and type seems inconsistent with your overall approach. Since you leverage motion priors for other components of the pipeline, why not also predict joint number and type using motion priors?

4. Lack of video demonstrations: Since this work focuses on modeling articulated objects from videos, video results showing temporal consistency and motion would strengthen the demonstration. Static images alone do not fully showcase the method's capabilities.

5. Missing texture visualization: Given that you use Gaussian Splatting as your representation with rendering loss for supervision, I would expect to see textured results in your visualizations.

**Questions:**

Please see the weaknesses. I am open to raising my score if my concerns are addressed.

---

> ### Author Response · Authors · 2025-11-21
> **Response to The Reviewer**
>
> We appreciate your acknowledgement of the effectiveness and technical contribution of our method. Please refer to the general response for a summary of all changes we have made in this revision. We address your concerns as follows:
>
> 1. **VideoArtGS-20 as a contribution.**
>
> We wish to clarify that the VideoArtGS-20 dataset itself is not claimed as a contribution. Our contributions, as stated in the paper, are:
> - The VideoArtGS method, which achieves state-of-the-art performance.
> - Motion prior guidance framework and hybrid assignment module
> - Comprehensive experimental validation and ablation studies
>
> VideoArtGS-20 serves as a **benchmark** to validate our method on complex multi-part objects. The contribution lies in the rigorous evaluation protocol and the adaptation of baselines to this challenging scenario, rather than the data generation process itself.
>
> 2. **Reconstruction smoothness.**
> We acknowledge that monocular video reconstruction may exhibit lower surface smoothness compared to static multi-view methods. However, this comparison must be contextualized:
> - **Fundamental Difficulty**: Monocular reconstruction is an ill-posed problem with inherent ambiguities due to limited viewpoint coverage, a challenge distinct from multi-view setups.
> - **State-of-the-Art Performance**: Despite these constraints, our method substantially outperforms all existing monocular baselines in terms of geometric fidelity and kinematics accuracy, as noted by other reviewers. We believe the fair comparison is against these relevant monocular methods, where VideoArtGS demonstrates superior performance.
>
> 3. **GPT-4o for joint count and type prediction**
> - **Appropriate scope**: Instead of asking for precise geometric parameters, we restrict GPT-4o to predict only discrete structural priors (joint count and types). This strategy leverages the model's semantic understanding while avoiding the inaccuracies typically associated with LLM spatial reasoning.
> - **Robustness over Heuristics**: Compared to motion-based heuristics (e.g., [1] or ArtGS), which rely on handcrafted assumptions like motion thresholds, GPT-4o demonstrates superior generalization on in-the-wild data.
> - **Performance Analysis**: As detailed in Appendix C.5, this approach is highly effective. On Video2Articulation-S (73 videos), GPT-4o made only 2 errors (misclassified joint types due to subtle motion). On VideoArtGS-20, it had 3 errors (underestimating parts in complex objects). These failure cases are mostly limited to scenarios with ambiguous motion or extreme structural complexity.
>
> 4. **Video demonstrations and Texture visualization**
> We would like to clarify that the video demonstrations and numerous interactive textured mesh files were included in our original submission via the [anonymous project website](https://videoartgs-2026.github.io) (linked in the Abstract).
> We respectfully invite the reviewer to examine these materials, which include:
>
> - **Rendered Visualizations**: High-resolution renders explicitly demonstrating the temporal consistency of our method.
> - **Physics Integration**: Real-time interactive manipulation demonstrations within the IsaacSim environment.
>
> We hope our response has thoroughly addressed your concerns. We are open to further discussion regarding any remaining questions or concerns. If our response and additional results have addressed your concerns, we would greatly appreciate your consideration of a higher score. Your suggestions are instrumental in improving the quality of our paper, and we sincerely thank you for providing your valuable feedback.

---

### Official Review · Reviewer_b5x2 · 2025-10-27

**Soundness:** 3
**Presentation:** 3
**Contribution:** 3
**Rating:** 6
**Confidence:** 4

**Summary:**

The paper presents VideoArtGS, a method for reconstructing articulated 3D objects in a scene from videos. It starts from depth and motion estimation from VGGT and Tapip3d, and cluster points with similar motion pattern into parts. Then optimizes a gaussian deformation field with rendering and tracking losses. It achieves better results than prior works on the Video2Articulation dataset, and the VideoArtGS-20 dataset introduced by this paper.

**Strengths:**

- The method is sound. The classification of articulated motion into 4 predefined motion types makes monocular reconstruction more well-posed.
- It represent the motion of points as a mix of rigid transformation given the distance to each part, where the static parts are handled separately with "staticness" logit. This hybrid assignment is simple and seems effective for separating movable vs. base parts
- It introduces a new VideoArtGS-20 dataset and reports of large improvements over prior art
- Strong ablations showing the improvement from motion-prior init and other components.

**Weaknesses:**

- Contributions are slightly incremental. The core modeling designs are similar to ArtGS. Gaussians + deformation deformation fields is established in prior works; most novelty is in initialization of motion type and assignment of points to parts.
- Heavy reliance on upstream systems (VGGT, TAPIP3D, GPT-4o). This could make the pipeline complex and less scalable.

**Questions:**

- Can authors clarify the use of GPT-4o vs 3D tracking, in terms of identifying the motion type and clustering?
- The deformation in Eq(1) assigns a soft weighting to each point, while this makes sense from an optimization perspective, the motion most of the objects/furnitures shown in the paper are controlled by a single rigid transformation.
- How is the occluded part of the scene represented/reconstructed and what happens if some part of the geometry is missing in the input video? e.g., the back of the table
- Are there motion types that are excluded by the 4 motion types used in the paper, and if so, how to combine those?
- The reconstruction results look not as high quality as recent 3d object/part generation models. Potentially combining those would improve the results.

---

> ### Author Response · Authors · 2025-11-21
> **Response to The Reviewer (1 / 2)**
>
> We appreciate your positive assessment that our method is "sound" with "strong ablations" and noting our "large improvements over prior art." Please refer to the general response for a summary of all changes we have made in this revision. We address your concerns as follows:
>
> 1. **Incremental contributions.**
> We respectfully disagree with the characterization of our contributions as "incremental." While we build upon ArtGS's articulation framework, our key innovations are:
> - **Novel Motion Prior Guidance Pipeline**: We introduce a complete framework for analyzing, filtering, and clustering 3D tracks to initialize articulation parameters. This pipeline is entirely absent in ArtGS and is critical for solving the ambiguity inherent in monocular inputs.
> - **Hybrid Center-Grid Assignment**: We propose a novel module to resolve complex geometries that ArtGS fail to reconstruct accurately. This significantly improves topological consistency.
> - **Fundamental Shift to Monocular Video**: Most critically, ArtGS relies on multi-view, multi-state capture, which severely limits its real-world application. Our method unlocks the capability to reconstruct articulated objects from monocular video, which is a paradigm shift that makes the technology practical for widespread use.
>
> 2. **Reliance on upstream systems**
> - **VGGT for depth and pose estimation**
>     - **Standard Modular Paradigm**: Given monocular RGB input, pose and depth estimation are a prerequisite. We regard the use of upstream estimators as a standard practice, analogous to how static scene reconstruction methods universally rely on COLMAP without diminishing their specific contributions.
>     - **Estimator-Agnostic Framework**: We adopt VGGT from SpatialTrackerV2 merely as an off-the-shelf tool due to its robustness in dynamic scenes. However, our framework is agnostic to the specific upstream estimator and can flexibly incorporate any state-of-the-art method (e.g., MegaSaM or Depth-Anything-V3) that provides good estimation.
>
> - **TAPIP3D for motion priors.**
>     - **Rationale for 3D Motion Priors**: We posit that explicit motion priors are essential for disambiguating complex articulated objects from monocular video. We select a native 3D tracker (TAPIP3D) rather than lifting 2D tracks (e.g., CoTracker3) into 3D space. This decision is based on our observation that lifting 2D tracks introduces depth unprojection ambiguities, where errors and noise are significantly magnified.
> 	- **Tracker-Agnostic & Evolvable**: While TAPIP3D serves as a robust initialization, our framework is tracker-agnostic. It is designed to benefit seamlessly from the rapid advancements in pre-trained tracking models—as upstream trackers improve, our initialization improves naturally.
> 	- **Robustness & Refinement**: We treat the output of TAPIP3D as a noisy initialization rather than ground truth. A key contribution of our pipeline is the ability to filter noise and refine these tracks. Our learnable hybrid center-grid assignment module is optimized via gradient descent, allowing the network to dynamically correct tracking errors during reconstruction. As demonstrated in Appendix C.2, our method not only tolerates input noise but also effectively improves the track quality in the final output. Please refer to **Appendix C.2 for full results**.
>
> - **GPT-4o for part number and joint types.**
>     - **Appropriate scope**: Instead of asking for precise geometric parameters, we restrict GPT-4o to predict only discrete structural priors (joint count and types). This strategy leverages the model's semantic understanding while avoiding the inaccuracies typically associated with LLM spatial reasoning.
>     - **Robustness over Heuristics**: Compared to motion-based heuristics (e.g., [1] or ArtGS), which rely on handcrafted assumptions like motion thresholds, GPT-4o demonstrates superior generalization on in-the-wild data.
>     - **Performance Analysis**: As detailed in Appendix C.5, this approach is highly effective. On Video2Articulation-S (73 videos), GPT-4o made only 2 errors (misclassified joint types due to subtle motion). On VideoArtGS-20, it had 3 errors (underestimating parts in complex objects). These failure cases are mostly limited to scenarios with ambiguous motion or extreme structural complexity.
>
> - **Clarification on Complexity and Scalability.**
>     - **Modular Design Philosophy**: We clarify that our pipeline is designed as a modular framework rather than a monolithic system tied to specific models. Our pipeline adopts a "plug-and-play" modular design. Tools like VGGT, TAPIP3D, and GPT are instantiated components, not hard constraints.
>     - **Automatic Performance Gains**: A key advantage of this design is that our method naturally evolves with these foundation models. For instance, replacing VGGT with "Depth-Anything-V3" would immediately boost our reconstruction quality.
>
> [1] Building Rearticulable Models for Arbitrary 3D Objects from 4D Point Cloud

---

> ### Author Response · Authors · 2025-11-21
> **Response to The Reviewer (2 / 2)**
>
> 3. **GPT-4o vs. tracking for motion type identification**
> The role of GPT4o is to identify how many parts are moving and their types in the video (e.g., 4 revolute + 2 prismatic), and the tracking model's role is to provide motion priors for initialization (centers, joint parameters) and additional supervision (i.e., tracking loss).  In order to make better use of this motion prior, we will analyze and cluster the point trajectories output by the tracking model to obtain the initialization of part centers and joint parameters. This initialization has been proven to be extremely effective.
>
> 4. **Soft weighting in deformation**
> During training, we use soft assignment (Eq. 3) to enable gradient-based optimization. We gradually reduce the softmax temperature during training following ArtGS, making assignments increasingly sharp and approaching a one-hot distribution. At test time, Gaussians are effectively controlled by a single rigid transformation, ensuring physically valid articulation.
>
> 5. **Occluded geometry**
> You identify an important consideration. Our method handles partial occlusion as follows:
> - We initialize Gaussians from visible regions in the first N frames
> - For regions occluded during static frames but visible during motion (e.g., the interior of a drawer), Gaussians will continue to be desified when they appear,  adapting to the newly visible region during motion.
> - Completely occluded regions cannot be reconstructed without additional priors. Integration with generative models for occluded region completion is a valuable future direction.
>
> 6. **Motion types beyond the four classes**
> Our current framework focuses on static, prismatic, revolute, and noise—encompassing the vast majority of common articulated objects. These primitives are the fundamental "atomic" building blocks of kinematic chains in articulated objects. Complex motions like screw joints (combined rotation and translation) are an interesting future extension.
>
> 7. **Comparison with generation models**
> We agree this is an exciting direction. Generative models could provide shape priors for occluded regions or canonical geometry, while our method provides articulation parameters and motion-consistent reconstruction. A hybrid approach is worth exploring in future work.
>
> We hope our response has thoroughly addressed your concerns. We are open to further discussion regarding any remaining questions or concerns. If our response and additional results have addressed your concerns, we would greatly appreciate your consideration of a higher score. Your suggestions are instrumental in improving the quality of our paper, and we sincerely thank you for providing your valuable feedback.

---

### Official Review · Reviewer_L1iA · 2025-11-01

**Soundness:** 3
**Presentation:** 3
**Contribution:** 2
**Rating:** 6
**Confidence:** 3

**Summary:**

This paper presents a method for reconstructing articulated objects from a single monocular video. The authors address the  problem by using two main components: first, a motion prior guidance pipeline that analyzes and filters 3D tracks from an upstream model (TAPIP3D) to initialize articulation parameters, and second, a hybrid center-grid part assignment module to segment static bases from movable parts. The method utilizes an articulation-based deformation field over 3D Gaussian Splatting.

**Strengths:**

1. This work addresses the challenging problem of articulated object reconstruction from  monocular video.
2. The method achieves a significant reduction in error compared to baselines and demos are looking good.

**Weaknesses:**

1.  The pipeline's success is critically dependent on the quality of two separate pre-trained models: VGGT for depth/pose and TAPIP3D for 3D tracks. This is a "garbage in, garbage out" system, and failures in these upstream models will cascade.

2. Part segmentation is derived entirely from motion clustering. This will inherently fail for parts that are not moved in the video, parts with very subtle motion, or objects with many parts moving simultaneously. Or if the parts are not moved to its maximum extend. There could be more information like semantic priors to be utilized.

3. The method relies on GPT-4O to predict the number of joints and their types (revolute/prismatic). This is a significant external dependency and a potential point of failure that is not analyzed.

4. The motion pattern analysis (Sec 3.2) involves several components (RANSAC, SVD fitting) and thresholds. This pipeline may be brittle and highly tuned. Thus the generalization ability could be a concern.

5. On the new VideoArtGS-20 dataset, the authors compare against methods (like Video2Articulation) that were not designed for multi-part objects and had to be manually adapted, which may inflate the performance gap. Similar to the question1 I raised, I doubt the good performance is highly tuned with its own carefully contructed dataset and might not generalize well.

6. The method requires the video to begin with a static sequence to initialize the canonical 3D Gaussians. This is a major limitation that prevents its application to most in-the-wild videos.

**Questions:**

1. In real world, many articulated objects don't move themselves. The would be humans/robots operate them to open and close the prismatic and revolute joints. This would definately result in occlusion. In the demos, human operators carefully move the objects to prevent occlusion. But if we use web videos where people operates them normally, how would you consider the occlusion problems? As I mentioned in weaknesses 6, the method relies on static start, if the inner part of a prismatic part like a drawer was occluded in the dynamic part, the method would lilely to fail?

2. Since the demo results looks good, I will give my initial score as marginally above the acceptance threshold. But those weaknesses I listed I still got many concerns on the work. I will consider the reviews from other reviewers and might adjust my score if my concerns are not resolved.

---

> ### Author Response · Authors · 2025-11-21
> **Response to The Reviewer (1 / 3)**
>
> Thank you for acknowledging that our work "addresses the challenging problem of articulated object reconstruction from monocular video" with "significant reduction in error" and "good-looking demos." Please refer to the general response for a summary of all changes we have made in this revision. We address your concerns as follows:
>
> 1. **Dependency on upstream models**
> - **VGGT for depth and pose estimation**
>     - **Standard Modular Paradigm**: Given monocular RGB input, pose and depth estimation are a prerequisite. We regard the use of upstream estimators as a standard practice, analogous to how static scene reconstruction methods universally rely on COLMAP without diminishing their specific contributions.
>     - **Estimator-Agnostic Framework**: We adopt VGGT from SpatialTrackerV2 merely as an off-the-shelf tool due to its robustness in dynamic scenes. However, our framework is agnostic to the specific upstream estimator and can flexibly incorporate any state-of-the-art method (e.g., MegaSaM or Depth-Anything-V3) that provides good estimation.
>
> - **TAPIP3D for motion priors.**
>     - **Rationale for 3D Motion Priors**: We posit that explicit motion priors are essential for disambiguating complex articulated objects from monocular video. We select a native 3D tracker (TAPIP3D) rather than lifting 2D tracks (e.g., CoTracker3) into 3D space. This decision is based on our observation that lifting 2D tracks introduces depth unprojection ambiguities, where errors and noise are significantly magnified.
> 	- **Tracker-Agnostic & Evolvable**: While TAPIP3D serves as a robust initialization, our framework is tracker-agnostic. It is designed to benefit seamlessly from the rapid advancements in pre-trained tracking models—as upstream trackers improve, our initialization improves naturally.
> 	- **Robustness & Refinement**: We treat the output of TAPIP3D as a noisy initialization rather than ground truth. A key contribution of our pipeline is the ability to filter noise and refine these tracks. Our learnable hybrid center-grid assignment module is optimized via gradient descent, allowing the network to dynamically correct tracking errors during reconstruction. As demonstrated in Appendix C.2, our method not only tolerates input noise but also effectively improves the track quality in the final output.
>
> - **Clarification on Complexity and Scalability.**
>     - **Modular Design Philosophy**: We clarify that our pipeline is designed as a modular framework rather than a monolithic system tied to specific models. Our pipeline adopts a "plug-and-play" modular design. Tools like VGGT, TAPIP3D, and GPT are instantiated components, not hard constraints.
>     - **Automatic Performance Gains**: A key advantage of this design is that our method naturally evolves with these foundation models. For instance, replacing VGGT with "Depth-Anything-V3" would immediately boost our reconstruction quality.
>
> - **Regarding "garbage in, garbage out"**: Our method actively filters noise from tracking outputs, making noisy tracks into reliable initialization. Our hybrid center-grid assignment module is learnable—its parameters are optimized via gradient descent during training, which naturally corrects errors from tracking noise. We have added track metrics in **Appendix C.2** to demonstrate our method's track correction capability.
>
>
> | Metric | EPE↓ TAPIP3D | EPE↓ Ours | δ0.05↑ TAPIP3D | δ0.05↑ Ours | δ0.10 ↑ TAPIP3D | δ0.10↑ Ours |
> | :--- | :---: | :---: | :---: | :---: | :---: | :---: |
> | 100481 | 0.16 | **0.12** | 59.04 | **66.95** | 65.40 | **72.76** |
> | 101284 | 0.03 | **0.01** | 88.99 | **95.96** | 90.46 | **96.75** |
> | 101287 | 0.08 | **0.02** | 70.11 | **91.73** | 74.16 | **93.16** |
> | **Average** | 0.16 | **0.09** | 55.57 | **73.02** | 62.07 | **76.98** |
>
> We present a portion of the results here; please refer to **Appendix C.2 for full results and analysis**.
>
> 2. **Motion-based part segmentation.**
> You raise a valid concern about parts with subtle or no motion, which has been discussed in Appendix B. We clarify:
> - Our method is designed for videos where articulation motion occurs (as stated in our problem formulation). For parts that do not move in the video, even if we can segment them, we cannot infer their articulation parameters. This is an inherent characteristic of motion-based approaches.
> - Our method does not assume sequential part movement (although this is in line with the actual data capture situation). Our approach can handle scenarios where multiple parts move simultaneously.
> - Integration with semantic segmentation priors (e.g., SAM2, DINOv2) is indeed a promising future direction. Our current work establishes the motion-based framework as a foundation upon which such priors can be integrated.

---

> ### Author Response · Authors · 2025-11-21
> **Response to The Reviewer (2 / 3)**
>
> 3. **GPT-4o for joint prediction**
> - **Appropriate scope**: Instead of asking for precise geometric parameters, we restrict GPT-4o to predict only discrete structural priors (joint count and types). This strategy leverages the model's semantic understanding while avoiding the inaccuracies typically associated with LLM spatial reasoning.
> - **Robustness over Heuristics**: Compared to motion-based heuristics (e.g., [1] or ArtGS), which rely on handcrafted assumptions like motion thresholds, GPT-4o demonstrates superior generalization on in-the-wild data.
> - **Performance Analysis**: As detailed in Appendix C.5, this approach is highly effective. On Video2Articulation-S (73 videos), GPT-4o made only 2 errors (misclassified joint types due to subtle motion). On VideoArtGS-20, it had 3 errors (underestimating parts in complex objects). These failure cases are mostly limited to scenarios with ambiguous motion or extreme structural complexity.
>
> | Dataset | Object ID | Category | GT Joint | Pred. Joint |
> | :--- | :---: | :---: | :---: | :---: |
> | Video2Articulation-S | 19898 | Table | 1r | 1p |
> | Video2Articulation-S | 22433 | Table | 1r | 1p |
> | VideoArtGS-20 | 30666 | Table | 9p | 7p |
> | VideoArtGS-20 | 103811 | Printer | 6p | 1r1p |
> | VideoArtGS-20 | 1280 | Faucet | 2r | 3r |
>
> [1] Building Rearticulable Models for Arbitrary 3D Objects from 4D Point Cloud
>
>
> 4. **Motion pattern analysis robustness**
> Our method uses equal thresholds for line/circle fitting ($\epsilon_l/\epsilon_c$). Throughout our main experiments, we adopt $\epsilon_l=\epsilon_c=0.01$.
> - **Optimization over Filtering**: A key strength of our method is that it does not rely on perfect hard-coded filtering. Our method does not require filtering out all noise. The thresholds serve only to provide a reasonable initialization. Our learnable hybrid center-grid assignment module is optimized via gradient descent, allowing the system to naturally correct initialization errors and handle tracking noise during the training process.
> - **Robustness across variations**: As demonstrated in Tab. A.5 and Tab. A.6, our method exhibits high stability across a wide range of threshold values. The variations in final optimized metrics are minimal, confirming that our default choice ($\epsilon_l=\epsilon_c=0.01$) is not a brittle "magic number." Our experiments demonstrate robust performance across reasonable threshold variations.
>
> | $\epsilon_l/\epsilon_c$ | CD_s (V2A) | CD_s (0.100) | CD_s (0.010) | CD_s (0.005) | CD_m (V2A) | CD_m (0.100) | CD_m (0.010) | CD_m (0.005) | Axis (V2A) | Axis (0.100) | Axis (0.010) | Axis (0.005) | Pos (V2A) | Pos (0.100) | Pos (0.010) | Pos (0.005) |
> | :--- | :---: | :---: | :---: | :---: | :---: | :---: | :---: | :---: | :---: | :---: | :---: | :---: | :---: | :---: | :---: | :---: |
> | 100481 | 2.63 | **0.11** | **0.11** | 0.37 | 1.43 | **0.02** | 0.03 | 14.07 | 43.93 | **0.01** | 0.06 | 12.37 | 35.92 | 0.02 | **0.00** | 2.53 |
> | 101284 | 7.28 | **0.01** | **0.01** | **0.01** | 76.58 | **0.00** | **0.00** | **0.00** | 14.91 | **0.14** | 0.18 | 0.35 | 29.31 | **0.10** | 0.13 | 0.15 |
> | 101287 | 10.16 | **0.01** | **0.01** | **0.01** | 24.70 | **0.00** | **0.00** | **0.00** | 9.07 | 0.25 | **0.20** | 0.28 | 28.55 | 0.06 | **0.03** | 0.07 |
> | **Average** | 10.22 | 0.26 | **0.24** | 0.41 | 30.63 | 1.17 | **0.26** | 14.57 | 48.88 | 2.32 | **0.34** | 5.13 | 37.04 | 1.06 | **0.10** | 2.05 |
>
> We present a portion of the results here; please refer to **Appendix C.3 for full results and analysis**.
>
> 5. **Performance on VideoArtGS-20**
> We respectfully disagree that the performance gap is artificially inflated:
> - Our method also achieves very large improvements on the existing Video2Articulation-S benchmark, which contains 73 test videos (Table 1), demonstrating that our gains are not merely dataset-specific
> - Video2Articulation was designed for single-part objects, and we adapted it fairly for multi-part scenarios by segmenting video clips. VideoArtGS-20 features larger camera motion and part dynamics than Video2Articulation-S, which is the main reason for Video2Articulation's poor performance. This also demonstrates the importance of motion prior for reconstructing articulated objects from monocular video.

---

> ### Author Response · Authors · 2025-11-21
> **Response to The Reviewer (3 / 3)**
>
> 6. **Static initialization requirement**
> - **A Necessary Trade-off for Quality**: We acknowledge this assumption (discussed in Appendix B). However, we argue that this is a strategic trade-off. Monocular articulated reconstruction is fundamentally ill-posed: disentangling complex articulation from rigid motion and shape deformation without multi-view cues is extremely difficult. A brief static initialization allows for a robust canonical geometry estimation, which serves as the foundation for high-fidelity tracking in subsequent frames.
> - **Practical applicability.** Our method is already applicable to self-captured videos where users can easily provide a static initialization, covering substantial use cases in robotics and AR applications.
> - **Comparable to prior work.** Previous methods (Video2Articulation, RSRD) make similar or even stronger assumptions about input constraints.
> - **Future Directions**: We have identified this as a key area for future work and believe integrating generative priors can relax this requirement.
>
> 7. **Occlusion handling.**
> You identify an important consideration. Our method handles partial occlusion as follows:
> - We initialize Gaussians from visible regions in the first N frames.
> - For regions occluded during static frames but visible during motion (e.g., the interior of a drawer), Gaussians will continue to be desified when they appear,  adapting to the newly visible region during motion.
> - Completely occluded regions cannot be reconstructed without additional priors. Integration with generative models for occluded region completion is a valuable future direction.
>
> We hope our response has thoroughly addressed your concerns. We are open to further discussion regarding any remaining questions or concerns. If our response and additional results have addressed your concerns, we would greatly appreciate your consideration of a higher score. Your suggestions are instrumental in improving the quality of our paper, and we sincerely thank you for providing your valuable feedback.

---

### Official Review · Reviewer_4fq7 · 2025-11-01

**Soundness:** 2
**Presentation:** 2
**Contribution:** 2
**Rating:** 4
**Confidence:** 3

**Summary:**

This paper introduces VideoArtGS, a system for reconstructing articulated object digital twins—including geometry, segmentation, and joint parameters—from monocular videos. The method builds upon Gaussian Splatting frameworks and integrates 3D motion priors and hybrid part assignments to recover both geometry and kinematics.

Motion Prior Guidance: It uses precomputed 3D trajectories (via TAPIP3D) and adaptive fitting (line/circle) to distinguish prismatic vs. revolute joints. The fitted parameters (axis, anchor, and joint state) are optimized using bidirectional trajectory consistency (c2o and o2o) to ensure temporal coherence.

Hybrid Center–Grid Assignment: The paper proposes a mixed scheme that assigns Gaussians to movable parts based on a center–Mahalanobis distance metric, and to static regions using a voxel-hash grid.

Optimization: The framework assumes the first N frames are static for canonical initialization, then jointly optimizes geometry and motion over the entire sequence using a photometric and consistency loss.

Experiments on Video2Articulation-S and a newly introduced VideoArtGS-20 dataset show strong improvements over Video2Articulation and ArticulateAnything, both quantitatively (two orders of magnitude reduction in axis and position errors) and qualitatively (cleaner reconstructions and more realistic part motion).

**Strengths:**

Addresses an important real-world problem: reconstructing interactable articulated objects from monocular video.

Integrative framework: elegantly combines motion priors, kinematic reasoning, and differentiable Gaussian rendering.

Solid empirical improvements: consistent quantitative and qualitative gains over existing baselines.

Ablation studies (motion prior and hybrid assignment removal) are provided and clearly demonstrate their necessity.

The method seems reproducible in principle, suggesting strong implementation work.

**Weaknesses:**

Over-reliance on upstream priors: The system’s success depends on TAPIP3D and VGGT, yet no robustness or substitution tests are conducted (e.g., replacing with weaker or noisier trackers).

Unrealistic static-frame assumption: The model presumes the first N frames are motionless; this limits generalization to natural videos. No adaptive mechanism or failure handling is proposed.

Limited generalization evidence: The “two orders of magnitude” improvement is mainly on synthetic or semi-synthetic datasets; real-world, unconstrained sequences are underrepresented.

Insufficient dataset transparency: Details of VideoArtGS-20’s annotation, coordinate conventions, and release availability are not discussed.

Lack of failure analysis: No qualitative examples of difficult cases (occlusion, partial visibility, transparent parts, trajectory drift) are shown.

Unreported runtime and efficiency: The training time, memory footprint, and inference speed are not compared with prior works.

Terminology and claims: The term “digital twin” suggests full physical consistency, but the framework models only geometric and kinematic properties, not physical constraints.

**Questions:**

How does performance change if TAPIP3D is replaced or degraded? Please provide robustness tests under noisy trajectories or alternative trackers.

Can the model relax the static N-frame assumption? For example, by learning canonical frames adaptively or detecting stationary segments automatically?

How are prismatic vs. revolute joints differentiated numerically—what thresholds or heuristics are used for line/circle fitting, and how sensitive are results to them?

How are mixed center–grid weights tuned? Are they learned jointly or fixed heuristically?

Please provide runtime and GPU memory comparisons with Video2Articulation and ArticulateAnything.

Are the reported “two orders of magnitude” improvements averaged across all categories, or do some categories dominate?

Will VideoArtGS-20 be publicly released with ground-truth joint annotations and evaluation code? If not, how can others reproduce the reported metrics?

How does the method handle complex real-world artifacts (e.g., flexible cables, soft hinges) that break rigid joint assumptions?

---

> ### Author Response · Authors · 2025-11-21
> **Response to the Reviewer (1 / 3)**
>
> We appreciate your recognition of our work as addressing "an important real-world problem" with "solid empirical improvements" and "strong implementation work." Please refer to the general response for a summary of all changes we have made in this revision. We address your concerns as follows:
>
> 1. **Reliance on upstream priors (VGGT, TAPIP3D, GPT-4o)**
> - **VGGT for depth and pose estimation**
>     - **Standard Modular Paradigm**: Given monocular RGB input, pose and depth estimation are a prerequisite. We regard the use of upstream estimators as a standard practice, analogous to how static scene reconstruction methods universally rely on COLMAP without diminishing their specific contributions.
>     - **Estimator-Agnostic Framework**: We adopt VGGT from SpatialTrackerV2 merely as an off-the-shelf tool due to its robustness in dynamic scenes. However, our framework is agnostic to the specific upstream estimator and can flexibly incorporate any state-of-the-art method (e.g., MegaSaM or Depth-Anything-V3) that provides good estimation.
>
> - **TAPIP3D for motion priors.**
>     - **Rationale for 3D Motion Priors**: We posit that explicit motion priors are essential for disambiguating complex articulated objects from monocular video. We select a native 3D tracker (TAPIP3D) rather than lifting 2D tracks (e.g., CoTracker3) into 3D space. This decision is based on our observation that lifting 2D tracks introduces depth unprojection ambiguities, where errors and noise are significantly magnified.
> 	- **Tracker-Agnostic & Evolvable**: While TAPIP3D serves as a robust initialization, our framework is tracker-agnostic. It is designed to benefit seamlessly from the rapid advancements in pre-trained tracking models—as upstream trackers improve, our initialization improves naturally.
> 	- **Robustness & Refinement**: We treat the output of TAPIP3D as a noisy initialization rather than ground truth. A key contribution of our pipeline is the ability to filter noise and refine these tracks. Our learnable hybrid center-grid assignment module is optimized via gradient descent, allowing the network to dynamically correct tracking errors during reconstruction. As demonstrated in Appendix C.2, our method not only tolerates input noise but also effectively improves the track quality in the final output.
>
> | Metric | EPE↓ TAPIP3D | EPE↓ Ours | δ0.05↑ TAPIP3D | δ0.05↑ Ours | δ0.10 ↑ TAPIP3D | δ0.10↑ Ours |
> | :--- | :---: | :---: | :---: | :---: | :---: | :---: |
> | 100481 | 0.16 | **0.12** | 59.04 | **66.95** | 65.40 | **72.76** |
> | 101284 | 0.03 | **0.01** | 88.99 | **95.96** | 90.46 | **96.75** |
> | 101287 | 0.08 | **0.02** | 70.11 | **91.73** | 74.16 | **93.16** |
> | **Average** | 0.16 | **0.09** | 55.57 | **73.02** | 62.07 | **76.98** |
>
> We present a portion of the results here; please refer to **Appendix C.2 for full results**.
>
> - **GPT-4o for part number and joint types.**
>     - **Appropriate scope**: Instead of asking for precise geometric parameters, we restrict GPT-4o to predict only discrete structural priors (joint count and types). This strategy leverages the model's semantic understanding while avoiding the inaccuracies typically associated with LLM spatial reasoning.
>     - **Robustness over Heuristics**: Compared to motion-based heuristics (e.g., [1] or ArtGS), which rely on handcrafted assumptions like motion thresholds, GPT-4o demonstrates superior generalization on in-the-wild data.
>     - **Performance Analysis**: As detailed in Appendix C.5, this approach is highly effective. On Video2Articulation-S (73 videos), GPT-4o made only 2 errors (misclassified joint types due to subtle motion). On VideoArtGS-20, it had 3 errors (underestimating parts in complex objects). These failure cases are mostly limited to scenarios with ambiguous motion or extreme structural complexity.
>
> - **Clarification on Complexity and Scalability.**
>     - **Modular Design Philosophy**: We clarify that our pipeline is designed as a modular framework rather than a monolithic system tied to specific models. Our pipeline adopts a "plug-and-play" modular design. Tools like VGGT, TAPIP3D, and GPT are instantiated components, not hard constraints.
>     - **Automatic Performance Gains**: A key advantage of this design is that our method naturally evolves with these foundation models. For instance, replacing VGGT with "Depth-Anything-V3" would immediately boost our reconstruction quality.
> [1] Building Rearticulable Models for Arbitrary 3D Objects from 4D Point Cloud.

---

> ### Author Response · Authors · 2025-11-21
> **Response to the Reviewer (2 / 3)**
>
> 2. **Static-frame assumption**
> - **A Necessary Trade-off for Quality.** We acknowledge this assumption (discussed in Appendix B). However, we argue that this is a strategic trade-off. Monocular articulated reconstruction is fundamentally ill-posed: disentangling complex articulation from rigid motion and shape deformation without multi-view cues is extremely difficult. A brief static initialization allows for a robust canonical geometry estimation, which serves as the foundation for high-fidelity tracking in subsequent frames.
> - **Practical applicability.** Our method is already applicable to self-captured videos where users can easily provide a static initialization, covering substantial use cases in robotics and AR applications.
> - **Comparable to prior work.** Previous methods (Video2Articulation, RSRD) make similar or even stronger assumptions about input constraints.
> - **Future Directions.** We have identified this as a key area for future work and believe that integrating generative priors can relax this requirement.
>
> 3. **Real-world evaluation.** We have included extensive real-world qualitative results (Fig.4) demonstrating our method's effectiveness across diverse object categories. Quantitative evaluation on real-world data is challenging due to the lack of ground-truth joint parameters and geometry. Our synthetic benchmarks provide rigorous quantitative validation, while real-world results demonstrate practical applicability.
>
> 4. **Dataset transparency.**
> We provide comprehensive details of the VideoArtGS-20 dataset in Appendix A.1, including: complete dataset configuration (Table A.1) and visualization of all objects (Figure A.1)
> Our dataset composition: images, depths, camera poses, ground-truth part meshes, and joint parameters. We commit to releasing all data, processing scripts, and model checkpoints upon acceptance to ensure full reproducibility.
>
> 5. **Failure analysis.** We provide more visualization and analysis of failure cases in **Appendix C.5**, including GPT-4o prediction failure, imperfect part segmentation, and clustering error.
>     - **Imperfect Part Segmentation (Fig.A.3a)**. Our method relies primarily on motion cues. Consequently, when multiple spatially adjacent components exhibit identical motion patterns (e.g., parallel prismatic joints moving in unison), the method may fail to separate them. This is a geometric ambiguity that motion cues alone cannot resolve. As discussed in Appendix B, incorporating semantic priors (e.g., SAM) is a promising direction to address this.
>     - **Clustering Error (Fig.A.3b)**. In scenarios with high part complexity and significant tracking noise, the initial center estimation may be inaccurate. While our optimization is robust, excessively large initialization errors can lead the model to converge to local minima. We identify this as a limitation of unsupervised clustering in high-noise regimes, which could be mitigated by semantic information.
>
> 6. **Runtime and efficiency**
>
> | Method | GPU Memory | Runtime |
> | :--- | :---: | :--- |
> | ArticulateAnything | N/A | 2-5 minutes |
> | Video2Articulation | 24GB | 20-30 minutes per joint |
> | Ours | 12GB | 15-40 minutes |
>
> - We provide a detailed evaluation of runtime and memory usage on a single NVIDIA RTX 3090 in Appendix C.4 and Table A.7.
> - **Comparison with ArticulateAnything**: While ArticulateAnything has lower latency, it relies on GPT for parameter prediction rather than performing geometric reconstruction from video evidence. Our method prioritizes physical fidelity and reconstruction quality, achieving a favorable balance between accuracy and computational cost.
> - **Comparison with Video2Articulation**: Our method demonstrates significantly higher efficiency and lower memory costs. Video2Articulation processes joints sequentially, requiring 20-30 minutes per joint. For multi-joint objects, this accumulates to several hours. In contrast, our method employs a joint optimization framework that reconstructs all articulated parts simultaneously, completing the entire process in 15-40 minutes depending on part complexity.
>
> 7. **Term about "Digital Twin"**
> This term began from Ditto[2] or even earlier papers. Physical constraints are indeed a part that a digital twin should incorporate, but restoring Physical constraints is still a long way off at present, because restoring geometry and kinematics is already a huge challenge.
>
> [2] Ditto: Building Digital Twins of Articulated Objects from Interaction.

---

> ### Author Response · Authors · 2025-11-21
> **Response to The Reviewer (3 / 3)**
>
> 8. **Exchange tracker**.
> As we have discussed, we choose TAPIP3D because we believe that directly tracking points in 3D space is better than lifting 2D tracks (e.g., from CoTracker3) into 3D, where errors and noise are significantly magnified. While 2D tracks could provide additional supervision, they are difficult to analyze reliably in 3D space. Given the rapid progress in reconstruction and tracking domains, future models will quickly improve. Our framework can seamlessly integrate stronger models as they emerge. Testing with intentionally degraded models would not reflect realistic use cases or future applicability.
>
> 9. **Threshold sensitivity.**
> Our method uses equal thresholds for line/circle fitting ($\epsilon_l/\epsilon_c$). Throughout our main experiments, we adopt $\epsilon_l=\epsilon_c=0.01$.
>
> - **Optimization over Filtering**: A key strength of our method is that it does not rely on perfect hard-coded filtering. Our method does not require filtering out all noise. The thresholds serve only to provide a reasonable initialization. Our learnable hybrid center-grid assignment module is optimized via gradient descent, allowing the system to naturally correct initialization errors and handle tracking noise during the training process.
> - **Robustness across variations**: As demonstrated in Tab. A.5 and Tab. A.6, our method exhibits high stability across a wide range of threshold values. The variations in final optimized metrics are minimal, confirming that our default choice ($\epsilon_l=\epsilon_c=0.01$) is not a brittle "magic number." Our experiments demonstrate robust performance across reasonable threshold variations.
>
> 10. **Mixed-center grid.**
> Both the centers and the grid are **learnable parameters** and will be updated through gradient descent during the optimization process. Therefore, the error caused by the noise output of the tracking model can be corrected by rendering loss.
>
> 11. **Cross-category performance**
>
> | Method | CD-m V2A | CD-m Ours | Axis V2A | Axis Ours | Position V2A | Position Ours |
> | :---: | :---: | :---: | :---: | :---: | :---: | :---: |
> | Box | 0.34 | **0.10** | 0.06 | **0.06** | 2.31 | **0.17** |
> | Dishwasher | 3.34 | **0.14** | 3.27 | **0.11** | 11.93 | **0.39** |
> | Laptop | 0.13 | **0.10** | 8.40 | **0.06** | 3.48 | **0.22** |
> | Microwave | 0.20 | **0.07** | 0.40 | **0.26** | 9.49 | **0.10** |
> | Refrigerator | 22.05 | **0.49** | 18.91 | **0.67** | 23.08 | **0.30** |
> | Scissors | 8.52 | **0.04** | 30.21 | **0.32** | 5.86 | **0.23** |
> | Stapler | 51.26 | **0.19** | 46.07 | **0.05** | 50.19 | **1.83** |
> | StorageFurniture | 10.39 | **3.02** | 4.47 | **0.14** | 0.61 | **0.01** |
> | Table | 19.21 | **1.56** | 17.22 | **0.44** | 3.01 | **0.07** |
> | USB | 4.27 | **0.95** | 11.29 | **0.18** | 3.95 | **0.05** |
> | WashingMachine | 4.32 | **0.09** | 13.03 | **1.29** | 5.28 | **1.89** |
>
> We provide detailed results across different categories and objects on the Video2Articulation-S and VideoArtGS-20 dataset in Appendix C.1. We present a portion of the results on Video2Articulation-S here; please refer to Appendix C.1 for full results. We observe consistent improvements across all categories in both Video2Articulation-S and VideoArtGS-20 datasets.
>
> 12. **Dataset release.**
> We commit to releasing all data, processing scripts, and model checkpoints upon acceptance to ensure full reproducibility.
>
> 13. **Non-rigid joint.**
> Flexible objects can be modeled using MPM (Material Point Method), but it is not the focus of our current work.
>
> We hope our response has thoroughly addressed your concerns. We are open to further discussion regarding any remaining questions or concerns. If our response and additional results have addressed your concerns, we would greatly appreciate your consideration of a higher score. Your suggestions are instrumental in improving the quality of our paper, and we sincerely thank you for providing your valuable feedback.

---

### Author Response · Authors · 2025-11-21
**General Response (1 / 2)**

We sincerely thank reviewers 4fq7, L1iA, b5x2, and PkAR for their valuable feedback, especially highlighting our contributions on addressing an important challenging problem (4fq7, L1iA),  achieving large improvements over prior methods with strong ablations (4fq7, L1iA, b5x2), and effective technical designs (PkAR). We follow comments and suggestions from all reviewers and revise our manuscript (colored in blue) accordingly as follows:

1. **Reliance on upstream priors.** (Reviewer 4fq7, L1iA, b5x2)
- **VGGT for depth and pose estimation**
    - **Standard Modular Paradigm**: Given monocular RGB input, pose and depth estimation are a prerequisite. We regard the use of upstream estimators as a standard practice, analogous to how static scene reconstruction methods universally rely on COLMAP without diminishing their specific contributions.
    - **Estimator-Agnostic Framework**: We adopt VGGT from SpatialTrackerV2 merely as an off-the-shelf tool due to its robustness in dynamic scenes. However, our framework is agnostic to the specific upstream estimator and can flexibly incorporate any state-of-the-art method (e.g., MegaSaM or Depth-Anything-V3) that provides good estimation.

- **TAPIP3D for motion priors.**
    - **Rationale for 3D Motion Priors**: We posit that explicit motion priors are essential for disambiguating complex articulated objects from monocular video. We select a native 3D tracker (TAPIP3D) rather than lifting 2D tracks (e.g., CoTracker3) into 3D space. This decision is based on our observation that lifting 2D tracks introduces significantly magnified errors and noise.
	- **Tracker-Agnostic & Evolvable**: While TAPIP3D serves as a robust initialization, our framework is tracker-agnostic. It is designed to benefit seamlessly from the rapid advancements in pre-trained tracking models—as upstream trackers improve, our initialization improves naturally.
	- **Robustness & Refinement**: We treat the output of TAPIP3D as a noisy initialization rather than ground truth. A key contribution of our pipeline is the ability to filter noise and refine these tracks. Our learnable hybrid center-grid assignment module is optimized via gradient descent, allowing the network to dynamically correct tracking errors during reconstruction. As demonstrated in **Appendix C.2**, our method not only tolerates input noise but also effectively improves the track quality in the final output.

- **GPT-4o for part number and joint types.**
    - **Appropriate scope**: Instead of asking for precise geometric parameters, we restrict GPT-4o to predict only discrete structural priors (joint count and types). This strategy leverages the model's semantic understanding while avoiding the inaccuracies typically associated with LLM spatial reasoning.
    - **Robustness over Heuristics**: Compared to motion-based heuristics (e.g., [1] or ArtGS), which rely on handcrafted assumptions like motion thresholds, GPT-4o demonstrates superior generalization on in-the-wild data.
    - **Performance Analysis**: As detailed in **Appendix C.5**, this approach is highly effective. On Video2Articulation-S (73 videos), GPT-4o made only 2 errors (misclassified joint types due to subtle motion). On VideoArtGS-20, it had 3 errors (underestimating parts in complex objects). These failure cases are mostly limited to scenarios with ambiguous motion or extreme structural complexity.

- **Clarification on Complexity and Scalability.**
    - **Modular Design Philosophy**: We clarify that our pipeline is designed as a modular framework rather than a monolithic system tied to specific models. Our pipeline adopts a "plug-and-play" modular design. Tools like VGGT, TAPIP3D, and GPT are instantiated components, not hard constraints.
    - **Automatic Performance Gains**: A key advantage of this design is that our method naturally evolves with these foundation models. For instance, replacing VGGT with "Depth-Anything-V3" would immediately boost our reconstruction quality.

[1] Building Rearticulable Models for Arbitrary 3D Objects from 4D Point Cloud.

2. **Static-frame assumption**. (Reviewer 4fq7, L1iA)
- **A Necessary Trade-off for Quality**: We acknowledge this assumption (discussed in Appendix B). However, we argue that this is a strategic trade-off. Monocular articulated reconstruction is fundamentally ill-posed: disentangling complex articulation from rigid motion and shape deformation without multi-view cues is extremely difficult.
- **Practical applicability.** Our method is already applicable to self-captured videos where users can easily provide a static initialization, covering substantial use cases in robotics and AR applications.
- **Comparable to prior work.** Previous methods (Video2Articulation, RSRD) make similar or even stronger assumptions about input constraints.
- **Future Directions**: We have identified this as a key area for future work and believe integrating generative priors can relax this requirement.

---

> ### Author Response · Authors · 2025-11-21
> **General Response (2 / 2)**
>
> 3. **Sensitivity of fitting thresholds**. (Reviewer 4fq7, L1iA)
>
> Our method uses equal thresholds for line/circle fitting ($\epsilon_l/\epsilon_c$). Throughout our main experiments, we adopt $\epsilon_l=\epsilon_c=0.01$.
>
> - **Optimization over Filtering**: A key strength of our method is that it does not rely on perfect hard-coded filtering. Our method does not require filtering out all noise. The thresholds serve only to provide a reasonable initialization. Our learnable hybrid center-grid assignment module is optimized via gradient descent, allowing the system to naturally correct initialization errors.
> - **Robustness across variations**: As demonstrated in **Tab. A.5 and Tab. A.6**, our method exhibits high stability across a wide range of threshold values. The variations in final optimized metrics are minimal, confirming that our default choice ($\epsilon_l=\epsilon_c=0.01$) is not a brittle "magic number." Our experiments demonstrate robust performance across reasonable threshold variations. Please refer to **Appendix C.3 for full results and analysis**.
>
> 4. **Failure cases visualization and analysis.** (Reviewer 4fq7, L1iA)
>
> We provide more visualization and analysis of failure cases in **Appendix C.5**, including GPT-4o prediction failure, imperfect part segmentation, and clustering error.
> - **Imperfect Part Segmentation (Fig.A.3a)**. Our method relies primarily on motion cues. Consequently, when multiple spatially adjacent components exhibit identical motion patterns (e.g., parallel prismatic joints moving in unison), the method may fail to separate them. This is a geometric ambiguity that motion cues alone cannot resolve. As discussed in Appendix B, incorporating semantic priors (e.g., SAM) is a promising direction to address this.
> - **Clustering Error (Fig.A.3b)**. In scenarios with high part complexity and significant tracking noise, the initial center estimation may be inaccurate. While our optimization is robust, excessively large initialization errors can lead the model to converge to local minima. We identify this as a limitation of unsupervised clustering in high-noise regimes, which could be mitigated by semantic information.
>
> 5. **Detailed results on Video2Articulation-S and VideoArtGS-20 dataset.** (Reviewer 4fq7)
>
> We provide detailed results across different categories and objects on the Video2Articulation-S and VideoArtGS-20 dataset in **Appendix C.1**. We observe consistent improvements across all categories in both Video2Articulation-S and VideoArtGS-20 datasets.
>
> 6. **Comparison for gpu memory and run time.** (Reviewr 4fq7)
> - We provide a detailed evaluation of runtime and memory usage on a single NVIDIA RTX 3090 in **Appendix C.4 and Table A.7**.
> - **Comparison with ArticulateAnything**: While ArticulateAnything has lower latency, it relies on GPT for parameter prediction rather than performing geometric reconstruction from video evidence. Our method prioritizes physical fidelity and reconstruction quality, achieving a favorable balance between accuracy and computational cost.
> - **Comparison with Video2Articulation**: Our method demonstrates significantly higher efficiency and lower memory costs. Video2Articulation processes joints sequentially, requiring 20-30 minutes per joint. For multi-joint objects, this accumulates to several hours. In contrast, our method employs a joint optimization framework that reconstructs all articulated parts simultaneously, completing the entire process in 15-40 minutes depending on part complexity.

---

### Meta-Review · Area_Chair_eZYk · 2026-01-08

**Summary:**

This paper received mixed reviews. The main concerns raised in the reviews are:
1. heavy reliance on pretrained 3D and tracking models and LLMs with no robustness tests (`4fq7`, `L1iA`, `b5x2`).
2. crucial assumptions on the input video, including static capture in the beginning, simple motion, minimal occlusion by human operators; a complex pipeline with many carefully-tuned hyperparameters, which might be brittle and could limit generalization and scalability; limited evaluation on unconstrained, real-world scenes (`4fq7`, `L1iA`); inadequate discussion on failure cases (`4fq7`).
3. missing technical details (`4fq7`).
4. limited technical innovation (`b5x2`).
5. reconstructed geometry is coarse (`PkAR`).
6. absence of video results (`PkAR`).

Overall, the reviewer all recognizes the importance of the task but remain unconvinced about the robustness of the proposed approach given its heavy reliance on large pretrained models and a carefully tuned pipeline. The best way to dispel such doubts is to demonstrate many more diverse results alongside the input videos, and really show the scalability of the method. Based on the current state of the submission, I recommend Reject.

**Reviewer Concerns:**

- Concerns #1 and #2 are partially addressed by the clarification of the design choices and additional ablation experiments. However, robustness against potential errors of those upstream models as well as the hyperparameters on real-world application scenarios still remains questionable. The only way to dispel such doubts would be to provide a large collection of results to demonstrate the scalability of such an approach.
- I believe Concerns #3 and #6 have been addressed.
- Concerns #4 and #5 are still largely outstanding.

**Reviewer Scores:**

It's difficult to predict how the reviewers might update their rating as the main concerns revolve around the robustness of the proposed method on real-world scenarios, which is hard to argue against without showing many more real-world results.

---

### Decision · Program_Chairs · 2026-01-26

Reject